# WEPP: Phylogenetic placement achieves near-haplotype resolution in wastewater-based epidemiology

Pranav Gangwar[1], Pratik Katte[2], Manu Bhat[1], Yatish Turakhia[1]*

**1** Department of Electrical and Computer Engineering, University of California, San Diego, California, United States of America, **2** Department of Biomolecular Engineering, University of California, Santa Cruz, California, United States of America

* yturakhia@ucsd.edu

## Abstract

Wastewater-based epidemiology (WBE) is a cost-effective, unbiased, and time-efficient tool for public health surveillance. Although widely adopted since the COVID-19 pandemic, WBE remains underutilized in genomic epidemiology, as most tools are limited to lineage-level resolution and focus only on estimating lineage abundances from wastewater sequencing reads. Here, we present WEPP, a pathogen-agnostic pipeline that improves both the resolution and capabilities of WBE analysis. WEPP uses phylogenetic placement of sequencing reads onto mutation-annotated trees (MATs)—daily updated phylogenies of all globally available clinical sequences and their inferred ancestors—to sensitively and precisely identify a subset of haplotypes likely present in a sample. It also reports the abundance of each haplotype and lineage, and flags "unaccounted alleles"— those found in the sample but not explained by selected haplotypes—that may indicate novel variants. WEPP includes a powerful interactive dashboard for high-resolution visual analysis, allowing users to explore haplotype and lineage abundances, read-to-haplotype mappings, and unaccounted alleles within a global phylogenetic context. Applied to wastewater samples from multiple cities and pathogens, WEPP uncovered biological insights sometimes missed by other tools and enabled new WBE applications previously confined to clinical sequencing, such as identifying (i) intra-lineage haplotype clusters, (ii) multiple cluster introductions in a city, (iii) early haplotype detection up to five weeks before clinical confirmation, (iv) mutations from novel variants, and (v) circulating lineages missed by clinical surveillance. With these capabilities, WEPP can transform wastewater-based epidemiology into a more powerful tool for monitoring and managing infectious disease outbreaks.

**Data availability statement:** All relevant data are included within the manuscript and its Supporting Information files. The WEPP source code is publicly available under the MIT license at https://github.com/TurakhiaLab/WEPP, which also includes the dashboard codebase as a submodule hosted at https://github.com/pratikkatte/Wastewater-Dasboard. Comprehensive documentation for the tool is available at https://turakhia.ucsd.edu/WEPP to support new users.

**Funding:** This work was supported by funding from the U.S. Centers for Disease Control and Prevention through the Office of Advanced Molecular Detection (CDC contract #75D30123C17463) to YT, the Amazon Research Award (Fall 2022 CFP) to YT, and funding from the Hellman Fellowship given to YT. The funders provided support in the form of salaries for authors PG, PK, and YT, but had no role in study design, data collection and analysis, decision to publish, or preparation of the manuscript.

**Competing interests:** The authors have declared that no competing interests exist.

## Author summary

Since the COVID-19 pandemic, wastewater-based epidemiology (WBE)—which analyzes sewage samples to detect traces of pathogens circulating in a community—has been widely adopted worldwide for public health surveillance. While more cost- and time-efficient than clinical testing, WBE is complicated by the fact that sewage contains only small fragments from a wide mix of pathogen variants infecting the community. Consequently, existing WBE tools have been limited to coarse resolution at the strain or lineage level. Our tool, WEPP, addresses this challenge by harnessing the vast global repository of clinical genome sequences to identify near-exact matches in wastewater. This enhanced resolution enables powerful new WBE applications, such as detecting emerging undesignated variants or tracking the global spread of locally circulating variants using a comprehensive database—capabilities previously confined to clinical surveillance.

## Introduction

Wastewater carries a comprehensive mixture of pathogens shed by infected individuals in a catchment area, making wastewater-based epidemiology (WBE) a powerful tool for monitoring community-level pathogen dynamics that is also non-invasive, cost-effective, timely, and unbiased [1]. During the COVID-19 pandemic, WBE demonstrated successful applications in detecting emerging SARS-CoV-2 *lineages* (definition in Table 1) up to two weeks before clinical reporting [2–6], enabling timely public health interventions. This success has driven the widespread adoption of WBE across more than 55 countries [7], including many low- and middle-income countries (LMICs), with deployments ranging from university dormitories [2,8,9] to hospitals [10–12] and airports [13–15]. It has also catalyzed its expansion to several other pathogens, including influenza, respiratory syncytial virus (RSV), rhinovirus, monkeypox (Mpox), Zika, and Human papillomavirus (HPV) [16–19].

Despite immense success, substantial opportunities remain untapped in WBE, as, to date, most WBE tools have focused predominantly on estimating the proportions of various lineages [2,20–28]. As a result, several critical epidemiological applications that necessitate *haplotype-level resolution* (definition in Table 1) remain reliant on clinical sequencing. These applications include: (i) tracking intra-lineage haplotype clusters to study local transmission dynamics [29,30], (ii) identifying introductions of haplotype clusters into a region [31], and (iii) detecting emerging variants that are not yet designated with lineage labels [32,33]. Since clinical sequencing rates have been in sharp decline since the official end of the COVID-19 pandemic [34], it is imperative that we empower WBE with more capabilities, particularly by improving its resolution, as we transition to a broader post-pandemic surveillance era and expand WBE to more pathogens.

While there have been prior efforts [20] to reconstruct haplotypes directly from sequencing reads, these have only been applied to mixed samples containing

**Table 1. Glossary of terms.**

| Allele | It can refer to any variant of the nucleotide sequence at a particular genomic locus. In the context of WEPP, it refers to a single nucleotide at a specific genome site (e.g., 135T represents nucleotide T at site 135). |
|---|---|
| Haplotype | A complete set of alleles present in a viral genome, represented either by a clinically sequenced genome stored in the mutation-annotated tree (MAT) or by an inferred ancestral node within the MAT. WEPP selects from these haplotypes to explain amplicon sequencing data. |
| Lineage | A group of closely related haplotypes that share a common ancestor and characteristic mutations. Lineages are pre-annotated in the MAT. |

haplotypes from two very distinct clusters (separated by 200–400 mutations [20]), and they struggle to reconstruct haplotypes within clusters exhibiting low genetic diversity (separated by fewer than 100 mutations). As a result, these approaches are not well-suited for wastewater samples [35], which often feature complex mixtures of dozens of haplotypes with low genetic diversity. Their performance is further exacerbated when applied to short-read sequencing data, which remains the predominant sequencing technology in WBE applications [2,22,24–26,35]. Pipes et al. [36] proposed an alternative approach to overcome these challenges—rather than reconstructing haplotypes from scratch, they leveraged an expectation-maximization (EM) algorithm to select the most likely subset of haplotypes from a database of previously sequenced clinical samples, thus improving resolution over lineage abundance estimation tools. However, even this method is suited only for samples containing a relatively small number (around 10) of unique haplotypes [36], and the computationally intensive nature of the EM algorithm necessitates aggressive pre-filtering strategies that impact its haplotype selection accuracy (see Results). Moreover, this method does not discover novel or 'cryptic' variants [37–42] – variants that have not yet been observed in clinical sequencing.

To address these shortcomings, we introduce WEPP (Wastewater-based Epidemiology using Phylogenetic Placements)–a pathogen-agnostic pipeline that significantly improves the accuracy and resolution of wastewater-based epidemiology. WEPP places sequencing reads onto mutation-annotated trees (MATs) to identify likely haplotypes and estimate their relative abundances that best explain the sample. It also reports lineage abundances, 'unaccounted alleles' — *alleles* (definition in Table 1) present in wastewater but unexplained by selected haplotypes, along with parsimonious read-to-haplotype mappings. These unaccounted alleles closely resemble the 'cryptic' mutations described in prior studies [37,38,41,42], which are often indicative of a potentially novel circulating variant. However, unaccounted alleles may also arise from systematic sequencing or bioinformatic errors, or suboptimal haplotype selection (Results). To support deeper investigation, WEPP provides an interactive visualization dashboard that allows users to explore detected haplotypes within the context of a comprehensive phylogenetic tree and examine unaccounted alleles alongside both the supporting reads and their parsimony-based selected haplotypes. Such fine-grained wastewater analysis enables haplotype cluster tracking across locations and over time, as well as detecting novel variants through unaccounted alleles, which is not possible with current WBE tools or dashboards.

Through our extensive evaluation using diverse benchmarks across multiple pathogens, we demonstrate WEPP's excellent performance across a wide range of 54 simulated, 13 synthetic (laboratory-engineered), and 77 real-world wastewater samples chosen to reflect practical deployment scenarios. On simulated and synthetic SARS-CoV-2 wastewater samples, WEPP sensitively and precisely identified haplotypes that were, on average, within one single-nucleotide substitution of the true haplotypes, while significantly reducing the root-mean-square error of lineage-abundance estimates compared to existing methods. On real wastewater samples, WEPP exhibited higher concordance between lineage abundance estimates and clinical sequencing data. WEPP also accurately identifies intra-lineage haplotype clusters, detects the introduction of haplotype clusters into new regions ahead of their clinical confirmation, and flags unaccounted alleles associated with novel variants across both simulated and real-world datasets. In addition, WEPP generalizes

across sequencing technologies and pathogens, maintaining high accuracy in lineage abundance estimation on simulated data and identifying lineages in some real wastewater samples that were missed by clinical sequencing.

Overall, WEPP provides an automated, accurate, scalable, and flexible framework that enables near-haplotype-level resolution, thereby enabling timely interventions and expanding the scope of wastewater-based epidemiology for post-pandemic pathogen surveillance.

## Results

### WEPP overview

WEPP is a phylogeny-based pipeline for WBE that works complementarily with clinical sequencing efforts. It takes raw wastewater sequencing reads and a mutation-annotated tree (MAT)—comprehensive, up-to-date phylogenies of global clinical and inferred ancestral sequences maintained by the UShER toolkit [43,44]— as input and selects likely haplotypes corresponding to the tip and internal nodes of the MAT, respectively (Fig 1A). WEPP estimates the relative abundances of these haplotypes and their corresponding lineages, reports unaccounted alleles, and provides parsimonious read-to-haplotype mappings, all of which can be explored through an interactive dashboard (Fig 1B). These features extend the capabilities of WBE by: (i) detecting intra-lineage clusters circulating within a local catchment area, (ii) inferring introductions of new transmission clusters across regions, (iii) identifying unaccounted alleles that may indicate emerging variants, and (iv) performing detailed, read-level analysis of the wastewater sample.

Fig 1C provides a high-level overview of the WEPP's core algorithm, with a detailed description available in the Methods section. Briefly, WEPP starts by performing parsimonious placement of raw sequencing reads on the MAT, using an optimized version of the UShER algorithm tailored for the phylogenetic placement of sequencing reads, rather than whole-genome sequences (Methods). Based on the resulting placement distribution, WEPP scores and selects a subset of haplotypes, along with their nearest neighbors (around 5000 total haplotypes, Methods), to form a pool of candidate haplotypes. This pool is passed to a deconvolution algorithm, which is based on Freyja [2], to estimate the relative abundance of each haplotype. Since Freyja's formulation does not include a regularization term, to mitigate overfitting, WEPP only retains haplotypes above a user-defined abundance threshold (default: 0.5%) and iteratively refines this set. In each iteration, a new candidate pool is formed from the current haplotypes that are above this frequency threshold, along with their corresponding neighbors (default: 2 mutation radius; maximum 500 neighbors per haplotype), followed by deconvolution. This process continues until it reaches either convergence or a maximum iteration count (default: 10). WEPP applies an outlier detection algorithm to the residue of the deconvolution algorithm to flag unaccounted alleles when there is a significant gap between observed and estimated allele frequencies (Methods). It also reports uncertainty associated with each selected haplotype through the set of 'Uncertain Haplotypes' and the maximum single-nucleotide distance among them, which helps distinguish between uncertainty arising from identical haplotypes in the MAT from that introduced by sequencing reads not covering all sites in the genome (Methods). WEPP incorporates several optimizations to achieve pandemic-scale analysis, allowing it to place, in minutes to hours, millions of wastewater sequencing reads on SARS-CoV-2 MATs that consist of tens of millions of haplotypes and internal nodes (Methods).

WEPP's interactive dashboard can be launched as a local web server through WEPP's Snakemake pipeline (Methods) and is designed to enhance interpretability and support in-depth downstream analysis. As shown in Fig 1B(i), the dashboard presents a detailed phylogenetic view, similar to Taxonium [45], of all clinically sequenced haplotypes, with the colored circles corresponding to the haplotypes identified in the wastewater sample by WEPP. The dashboard displays the estimated abundances of all selected haplotypes, their associated lineages, along with their genome coverage calculated after parsimonious read assignment to selected haplotypes (Fig 1B(i)). Additionally, it displays uncertain haplotypes for each selected haplotype candidate, along with the maximum single-nucleotide distance among them, allowing users to know the reason behind the uncertain selection (Methods). The dashboard also shows the unaccounted alleles together with their residue, allele frequency, sequencing depth, and parsimonious haplotype candidates. Users can also perform

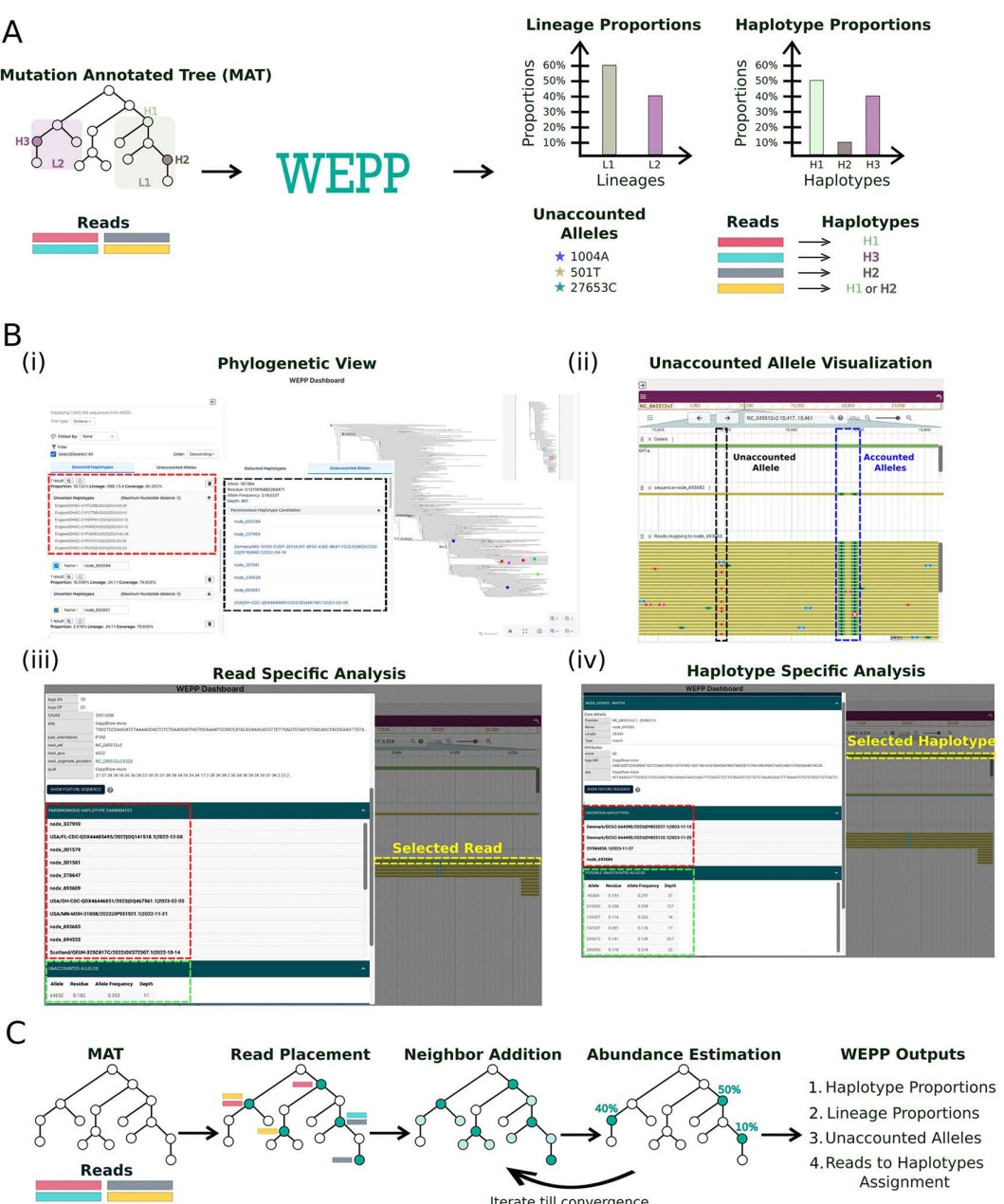

**Fig 1. Overview of the WEPP pipeline. (A)** WEPP input and output. **(B)** Features of the interactive WEPP Dashboard: **(i)** Phylogenetic view of the WEPP-inferred haplotypes, showing their estimated proportions, associated lineages, genome coverage by parsimonious read assignment, and uncertain haplotypes along with maximum nucleotide distance between them. It also displays unaccounted alleles with their residue, allele frequency, depth, and parsimonious haplotype candidates; **(ii)** Read analysis panel highlighting accounted and unaccounted alleles contained in reads mapped to a selected haplotype: the first highlighted site shows an alternate allele present in the reads but absent from the haplotype, while the second shows two alternate alleles present in both the reads and the haplotype; **(iii)** Read information panel displaying all parsimonious haplotype candidates, unaccounted alleles, and other read information like Equally Parsimonious Placements (EPP) for a selected read; **(iv)** Haplotype information panel listing the possible unaccounted alleles associated with the selected haplotype. **(C)** Key stages of WEPP's phylogenetic algorithm for haplotype detection and abundance estimation.

a detailed read-level analysis by selecting a haplotype (click on the MAT node) to view its characteristic mutations alongside those observed in the mapped reads (Fig 1B(ii)). Additional information about individual reads, or haplotypes, can be accessed by clicking on their corresponding objects, as shown in Fig 1B(iii) and Fig 1B(iv), respectively.

**WEPP significantly improves lineage abundance estimation accuracy and achieves near-haplotype resolution on simulated and synthetic wastewater data**

We evaluate WEPP's ability to estimate haplotype and lineage abundances using simulated and synthetic wastewater datasets with known ground truth (Methods). The synthetic wastewater samples were obtained from the Indiana Department of Health (IDOH) Laboratory and the data shared by Ferdous et al. [46], who sequenced their samples on Illumina and Oxford Nanopore Technologies (ONT) platforms, respectively.

The simulation analysis was performed on 10 samples using SWAMPy [47], a realistic wastewater simulation tool, in which the lineage abundances were made to resemble those reported by the CA-SEARCH surveillance efforts (https://searchcovid.info) between December 2022 and December 2023, and the haplotypes within those lineages were randomly chosen (Methods). To quantify the accuracy of lineage abundances and compare the results of WEPP to Freyja [2] and Pipes et al. [36], we used the Root Mean Square Error (RMSE) between expected and estimated abundances. To quantify the ability of WEPP to resolve haplotypes, we introduce two new metrics: (i) Weighted Haplotype Distance and (ii) Weighted Peak Distance (Methods). Briefly, Weighted Haplotype Distance quantifies how closely wastewater haplotypes match WEPP's inferred haplotypes (referred to as "peaks" to avoid confusion with wastewater haplotypes), where a lower value implies higher *sensitivity*. At the same time, Weighted Peak Distance assesses how well-estimated peaks align with true haplotypes in the sample, where a lower value implies higher *precision*. Since Freyja does not report actual haplotypes, we used root sequences corresponding to Freyja's reported lineages as their representative peaks for comparison.

We observed that WEPP significantly outperformed both Freyja and Pipes et al. on the most widely performed lineage abundance estimation task, achieving nearly 5-fold and 7.9-fold lower RMSE on average compared to Freyja and Pipes et al., respectively (Fig 2A). A more detailed sample-wise analysis (S1 Table) reveals that because Freyja only optimizes over phylogenetically-derived mutational barcodes for different lineages, it often struggles with neighboring lineages (e.g., EG.5 and EG.5.1 in the October 2023 sample, S1 Table), where a haplotype from a particular lineage circulating in the wastewater sample shares some mutations with its neighboring lineage. Similarly, WEPP significantly outperforms the technique proposed by Pipes et al., which is known to struggle with samples containing more than 10 haplotypes [36], whereas our simulated samples carried 100 haplotypes. Both Freyja and Pipes et al. report far more lineage than what is present in the wastewater sample (S1 Table). Overall, WEPP benefits not only from searching over all haplotypes in the MAT but also by including the inferred sequences at the internal nodes of the MAT, allowing it to more precisely identify the haplotypes circulating in the wastewater, thus leading to more reliable estimates. Moreover, WEPP exhibits strong sensitivity and precision in identifying haplotypes circulating in wastewater, achieving a 3.1-fold and 6.4-fold reduction in Weighted Haplotype Distance compared to Freyja and Pipes et al., respectively. Similarly, it attains a 6.9-fold and 7.5-fold lower Weighted Peak Distance relative to Freyja and Pipes et al. (Fig 2A and S1 Table). In fact, most of the WEPP distance values are close to zero, indicating that the WEPP-inferred haplotypes almost perfectly match the true haplotypes present in the sample. Although these datasets contained XBB-derived lineages present in every sample, to further evaluate WEPP's ability to identify *known* recombinant lineages, i.e., recombinants present in the MAT, we simulated two additional samples containing an Omicron-Omicron and an Omicron-Delta recombinant, respectively (Methods). For both samples, WEPP accurately recovered the proportions of all lineages and their corresponding haplotypes (Supplementary Results 3 in S1 Text and S12 Table), suggesting that WEPP maintains accuracy even in the presence of known recombinants.

We further evaluated all three tools over ten Illumina-sequenced synthetic control mixtures from the IDOH (Methods). Consistent with earlier findings, WEPP achieved substantial improvements over Freyja, with 4.9-fold lower lineage

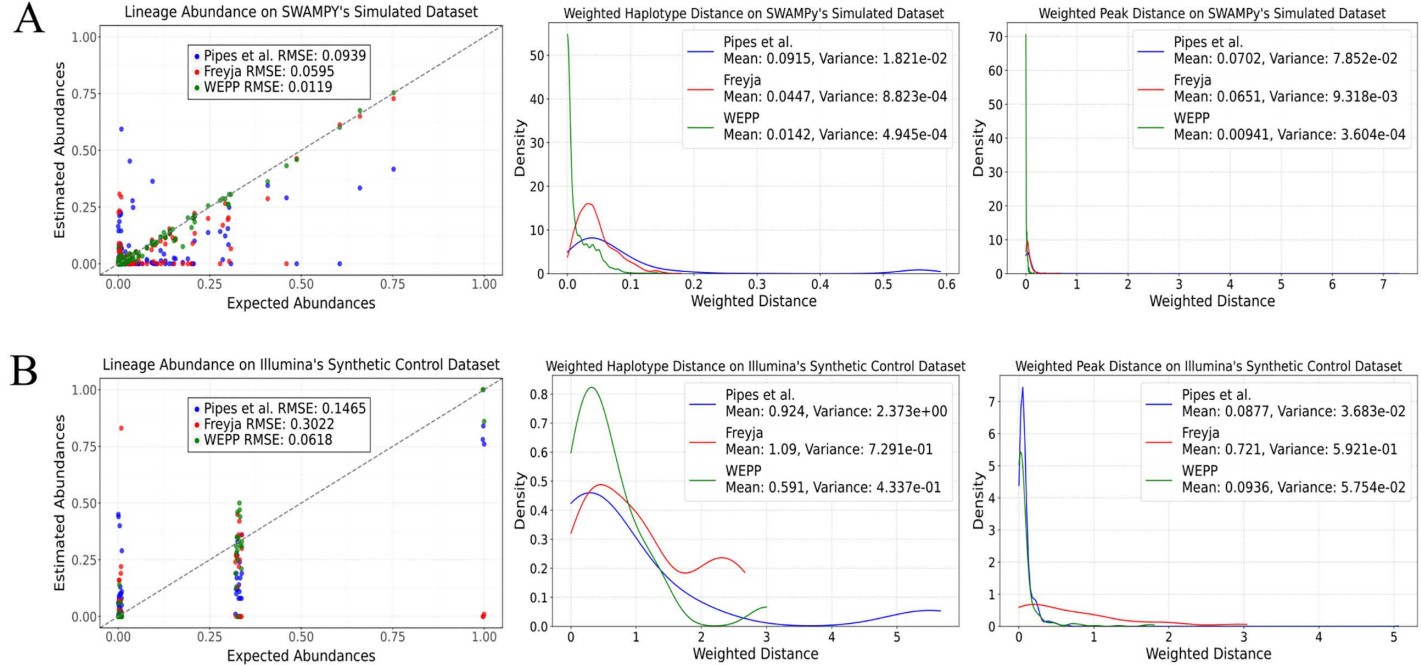

**Fig 2. Evaluation of lineage abundance accuracy and haplotype-level resolution in WEPP, Freyja, and Pipes et al. across simulated and synthetic control mixtures. (A)** Comparison of Lineage Abundance estimates, Weighted Haplotype Distance, and Weighted Peak Distance on SWAMPy simulated datasets. **(B)** Comparison of lineage abundance estimates, Weighted Haplotype Distance, and Weighted Peak Distance on the Illumina sequenced synthetic control datasets.

abundance RMSE, 1.8-fold lower Weighted Haplotype Distance, and 7.7-fold lower Weighted Peak Distance. The technique of Pipes et al. likely benefited from the simplicity of the synthetic samples, which included only 1–3 haplotypes, enabling their method to achieve a 1.1-fold lower Weighted Peak Distance than WEPP, indicating marginally better precision. However, WEPP performed significantly better on remaining metrics, achieving 2.4-fold and 1.6-fold reductions on RMSE and Weighted Haplotype Distance, respectively, compared to Pipes et al. (Fig 2B and S1 Table). Additionally, across both simulated and synthetic datasets, WEPP exhibited higher precision and sensitivity in lineage detection for lineages with over 0.5% abundance (S11 Table).

We observed similar trends when evaluating WBE tools on three synthetic wastewater samples from Ferdous et al. [46], sequenced using high-error long reads (ONT). WEPP achieved a 3.2-fold reduction in RMSE compared to Freyja (Fig A(i) in S1 Text and S1 Table, Supplementary Results 2 in S1 Text). Despite the elevated error rates in ONT reads, WEPP's inferred haplotypes remained close to the ground truth, highlighting its robustness and generalizability.

Overall, WEPP's strong performance on both simulated and synthetic wastewater samples across two distinct sequencing technologies underscores its ability to enhance the accuracy and resolution of wastewater surveillance.

## WEPP can accurately identify intra-lineage circulating clusters

The increased resolution of WEPP's analysis at the haplotype level has the potential to unlock powerful downstream applications for epidemiologists using wastewater data. During much of the COVID-19 pandemic, local transmission at any given time point was often driven by just a few dominant viral lineages [2,48–50]. These broad lineage-level trends could be effectively tracked using existing wastewater-based epidemiology (WBE) tools, such as Freyja. However, within many of these lineages lies substantial genetic diversity, e.g., two haplotypes belonging to the BA.1 lineage could differ by as

many as 50 substitutions (Fig B in S1 Text), which can mask important epidemiological dynamics. For example, distinct co-circulating clusters within a lineage may represent multiple independent introductions of the virus into a region or the emergence of localized outbreak events, and public health officials have typically needed to turn to clinical sequencing data to detect these patterns [51–55].

We evaluated WEPP's ability to detect intra-lineage haplotype clusters using simulated wastewater samples based on clinical SARS-CoV-2 sequences circulating in San Diego in January and September 2023. For each month, we used CA-SEARCH to identify all clinically observed haplotypes, simulated corresponding wastewater samples with SWAMPy, and compared WEPP's inferred haplotypes to the ground truth (Fig 3A–3B).

In September 2023, EG.5.1.4, XBB.1.5.10, and XBB.1.16.6 together comprised 45% of San Diego sequences, while in January 2023, BQ.1.1 and BQ.1 represented 24% (Fig 3A–3B). These lineages contained multiple genetically distinct haplotype clusters (e.g., BQ.1.1 clusters differ by up to 28 substitutions as shown in Fig B(ii) in S1 Text), which existing WBE tools, such as Freyja, cannot discern. WEPP not only produced more accurate lineage-level estimates (S2 Table) but also recovered all haplotypes circulating in September (Fig 3A) and most of the clusters circulating in January (Fig 3B).

By resolving this intra-lineage variation directly from wastewater samples, WEPP could provide a complementary lens for real-time surveillance and finer-scale tracking of pathogen spread, especially in settings with limited clinical sequencing capacity.

## WEPP can enable early detection of new cluster introductions from wastewater sequencing

Previous studies have demonstrated that wastewater surveillance can detect emerging viral lineages up to two weeks before clinical sequencing [2]. While this capability has underscored the utility of wastewater surveillance as an early warning system, current WBE tools cannot investigate the introductions and origins of emerging variant clusters in different geographical locations. Specifically, while tools like ClusterTracker [31] provide this functionality for clinical sequencing data and are widely used by public health officials in the US, we are not aware of any analogous capability for wastewater data.

Similar to ClusterTracker, WEPP's increased resolution has the potential to flag the introduction of new variant clusters in a geographical region and track their likely source of origin, but only using wastewater sequencing data. We used simulated data to evaluate this capability since such a level of ground truth is not available for any real-world data. Specifically, we simulated cluster emergence of two transmission events discovered by ClusterTracker: (1) New York to California transmission of an EG.5.1.9 variant cluster in September 2023 (Fig 3C), and (2) California to Delaware transmission of an HU.1.1 variant cluster in July 2023 (Fig 3D). Detailed methodology for simulating cluster emergence in wastewater surveillance is provided in Methods.

On both datasets, WEPP demonstrated promising results. Since the actual haplotypes that were introduced in the target region (California and Delaware, respectively) were removed from the input MAT, WEPP instead detected two closely related haplotypes at 4% and 1% abundance in both cases— belonging to (1) the New York originating cluster in the first experiment (Fig 3C), and (2) the California originating cluster in the second (Fig 3D). Moreover, in both cases, the haplotypes surrounding the WEPP's detected haplotypes belonged to their probable sources—New York and California, respectively—strengthening confidence in the inferred transmission pathways. These results underscore WEPP's potential to detect emerging haplotype clusters in a region and identify their possible sources of origin, possibly even before they are detected through clinical sequencing. This opens a new application for wastewater sequencing, thereby expanding its role in public health surveillance.

## WEPP identifies novel variants in simulated and real wastewater data

Timely detection of new variants is essential for guiding public health responses and can help save lives [56,57]. Delays in detecting emerging variants from clinical sequencing are well documented [58,59]; moreover, clinical sequencing rates

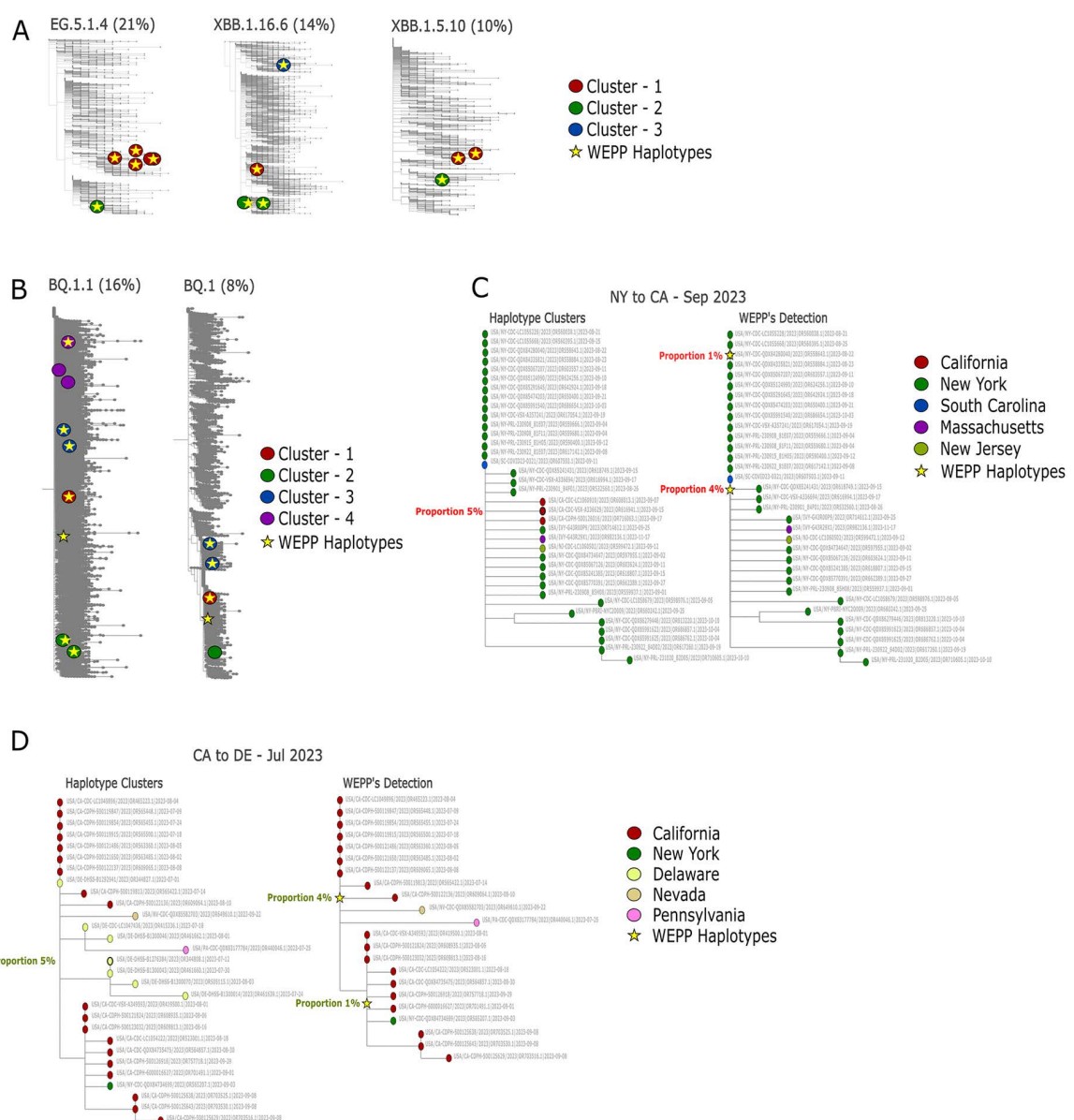

**Fig 3. Demonstrating WEPP's haplotype detection capabilities across various scenarios, with visualizations made with the help of WEPP's dashboard. (A)** WEPP's detection of intra-lineage haplotype clusters circulating in San Diego in September 2023. **(B)** WEPP's detection of intra-lineage haplotype clusters circulating in San Diego in January 2023. **(C)** Left: ClusterTracker indicates a likely variant cluster introduction from New York to California. Right: WEPP's detected haplotypes in the New York-origin cluster using a MAT excluding California haplotypes. **(D)** Left: ClusterTracker indicates a likely variant cluster introduction from California to Delaware. Right: WEPP's detected haplotypes in the California-origin cluster using a MAT excluding Delaware haplotypes.

declined sharply following the end of the Public Health Emergency declaration in May 2023 [34], which has impacted the ability to track new variants effectively. In contrast, due to its cost-effectiveness, wastewater surveillance programs remain active in many parts of the world and are therefore playing an increasingly important role in monitoring variants. While previous wastewater studies have identified early cryptic SARS-CoV-2 variants that were unexplained by clinical sequencing

[37–39], they have typically involved manual efforts and were limited in scope (e.g., focusing on rare, co-occurring mutations), which, as our results later indicate, could be prone to accuracy and sensitivity issues.

As described earlier, WEPP extends Freyja's deconvolution framework to identify alleles whose observed frequencies from wastewater deviate significantly from expectations based on the inferred haplotypes and their abundances. These deviations, referred to as unaccounted alleles, may signal cryptic mutations—potential early indicators of emerging variants or novel lineages. To assess the sensitivity and precision of this approach, we systematically varied the abundance of a novel haplotype and its genetic divergence from known sequences in the phylogenetic tree by removing its neighboring haplotypes (indicated by 'Node Removal Radius', Table 2). As evident in the results presented in Table 2, increasing the genetic divergence of the novel haplotype from known sequences makes it more challenging to detect its closest match in the MAT. This is reflected by the distance between the novel haplotype and the closest haplotype detected by WEPP, which exceeds the minimum value defined by the 'Node Removal Radius' at higher divergence. Nevertheless, WEPP consistently recovered all mutations associated with highly divergent haplotypes, even at low proportions. Moreover, nearly every unaccounted allele flagged by WEPP corresponded to 'Distance-Reducing Alleles' (Methods)—alleles that, if assigned to the correct inferred haplotype, would reduce dissimilarity to the true sequences in the sample. As the abundance of the novel haplotype increased, the number of unaccounted alleles declined, consistent with WEPP's adaptive outlier detection strategy that becomes more stringent in the presence of stronger signals (Methods). These findings highlight WEPP's ability to accurately detect private mutations and identify the closest phylogenetic matches for novel haplotypes, even when present at only 5% abundance. In addition, we evaluated WEPP's performance in the presence of a *novel* recombinant lineage, i.e., when the recombinant and its descendants were missing in the input MAT (Methods). In these samples, WEPP accurately selected the recombinant's two parental lineage haplotypes as candidates and flagged nearly all informative sites corresponding to the recombinant haplotype in unaccounted alleles (Supplementary Results 3 in S1 Text and S12 Table), which is evidence for the end users to infer the presence of a novel recombinant in the sample.

We further validated the capability of WEPP to identify novel variants by analyzing 22 SARS-CoV-2 wastewater samples from the Point Loma wastewater treatment plant (San Diego) between November 27, 2021, and February 7, 2022, and comparing them against 12,908 clinical sequences reported in San Diego during the peak surveillance period from November 2021 to March 2022 (S6 Table, Methods). WEPP identified 330 unaccounted alleles, 35.8% (118 alleles) of

**Table 2. Evaluation of WEPP's unaccounted alleles and closest detected haplotype in the presence of novel haplotypes.** WEPP's sensitivity and precision in detecting novel haplotypes are assessed by (1) varying the novel haplotype's proportion in the sample, and (2) adjusting the mutation distance between the novel haplotype and its closest sequence in the tree, defined by the 'Node Removal Radius'. The columns represent: (i) the novel haplotype's proportion, (ii) 'Node Removal Radius', (iii) the mutation distance between the novel haplotype and the closest haplotype detected by WEPP, (iv) the number of unaccounted alleles detected by WEPP, and (v) the number of alleles that reduce the distance between WEPP inferred haplotypes and the wastewater haplotypes.

Averaged results over four datasets

| Novel Haplotype Proportion | Node Removal Radius | Distance Between the Novel Haplotype and the Closest WEPP-Detected Haplotype | Unaccounted Alleles | Distance-Reducing Alleles |
|---|---|---|---|---|
| 20% | 3 | 3 | 3 | 3 |
| | 5 | 6.5 | 6.5 | 6.5 |
| | 7 | 8 | 8.25 | 8 |
| 10% | 3 | 3 | 8 | 8 |
| | 5 | 6.5 | 7.25 | 7.25 |
| | 7 | 7.5 | 8 | 8 |
| 5% | 3 | 3 | 9 | 8.25 |
| | 5 | 6.5 | 10.5 | 10 |
| | 7 | 8 | 10.5 | 10 |

which were completely absent from the clinical sequences, and an additional 43.0% (142 alleles) were extremely rare, present in fewer than 1% of clinical sequences. Importantly, 20 out of 22 datasets (90.1% datasets) contained unaccounted alleles not observed clinically in the city (S14 Table). According to prior definitions [38,39], these novel or clinically rare alleles would typically be categorized as 'cryptic mutations'. In other words, the unaccounted alleles identified by WEPP have a high overlap (nearly 79%) with the cryptic mutation definition, whose origins may stem from undetected low-prevalence variants or upstream processing artifacts. The remaining 21% unaccounted alleles, although present in clinical data, were flagged by WEPP because their observed frequencies in the wastewater sample deviated substantially from the expected frequencies based on haplotype selection, suggesting either the accumulation of previously observed mutations in circulating variants or systematic errors in sequencing or bioinformatics pipelines.

To evaluate WEPP's potential for cryptic lineage detection, we analyzed a subset of the datasets from Suarez et al. [37], who generated consensus sequences for several cryptic lineages. They identified "cryptic lineage-defining amino acid substitutions" (cryptic mutations) from prior studies and searched for sequencing reads containing multiple of these mutations, along with some additional filtering criteria. We analyzed a total of 104 cryptic mutations reported by Suarez et al. from 18 samples (S9 Table) and found that WEPP flagged 26% of them as unaccounted alleles. The remaining cryptic mutations were not flagged because 49% fell below WEPP's read-depth threshold, 20.2% fell below the depth-weighted allele-frequency cutoff, and 4.8% were already captured within WEPP's inferred haplotypes. In other words, mutations not flagged by WEPP did not carry a "strong" signal to be confidently classified as cryptic. These results underscore that because WEPP integrates global phylogenetic context alongside stringent allele frequency and sequencing depth thresholds, it ensures that only high-confidence cryptic mutations are reported as unaccounted alleles. While this approach may differ from earlier strategies, we believe it provides a principled and balanced framework for maximizing both sensitivity and specificity in detecting cryptic mutations.

Together, these findings demonstrate that by reporting unaccounted alleles, WEPP offers critical insights that can help public health officials assess emerging or anomalous patterns in pathogen evolution and uncover novel and cryptic variants with high precision and sensitivity.

### Reanalysis of real wastewater data with WEPP reveals intra-lineage clusters, multiple introductions, and concordance with later clinical sequences

We compared WEPP's lineage detection accuracy against Freyja using both short-read, low-error Illumina-sequenced wastewater samples from the Point Loma wastewater plant and long-read, high-error ONT-sequenced Columbia University's hospital wastewater surveillance data from Annavajhala et al. [10], with contemporaneous clinical sequencing data serving as a benchmark. Clinical sample collection was simultaneously conducted at both locations, which typically precedes public database deposition by a median of 10–63 days [60], but serves as a high-confidence "oracle" reference for our comparisons. Our first evaluation focuses on 22 Illumina-sequenced wastewater samples collected between November 27, 2021, and February 7, 2022, from the Point Loma Wastewater Treatment Plant, during a period of highest clinical sequencing efforts in the city. Additionally, we analyzed 13 long-read ONT-sequenced wastewater samples collected in December 2022 from Columbia University's hospital, described in Annavajhala et al., and compared them against 57 clinical sequences collected from the same hospital during that month. Lineage abundance estimates from wastewater were evaluated against clinical data to assess the sensitivity and concordance of both tools with circulating SARS-CoV-2 lineages.

A key distinction between the tools was Freyja's detection of a substantially greater number of lineages across both datasets (S3 and S4 Tables). However, many of these lineages were inferred at low relative abundances and lacked corresponding support from clinical sequencing, particularly in the San Diego data during the peak surveillance period (Fig 4 and S4 Table). As reflected in our controlled lineage abundance experiments (Fig 2), these lineages indicate a certain degree of overfitting in Freyja's estimations. To quantify the concordance between wastewater-inferred and clinical

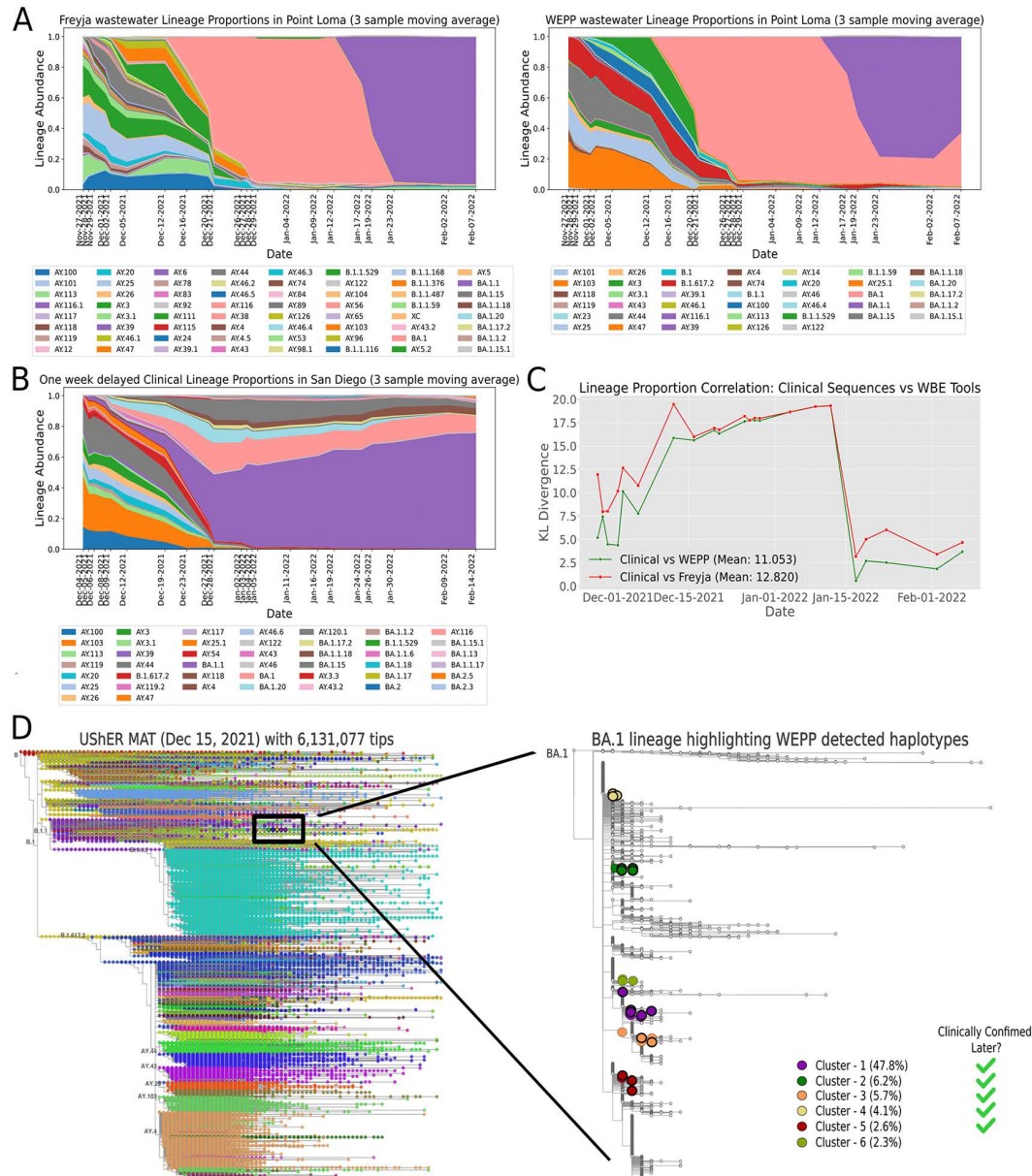

**Fig 4. Point Loma wastewater analysis with WEPP and Freyja. (A)** Lineage proportion estimates from Freyja (left) and WEPP (right) for Point Loma wastewater samples (Nov 27, 2021 – Feb 7, 2022). **(B)** Lineage proportions derived from one-week delayed "Oracle" clinical sequences in San Diego during the same period. **(C)** KL divergence between one-week delayed clinical sequence-derived lineage proportions and those estimated by WEPP and Freyja from wastewater. **(D)** (Left) Taxonium view of the UShER MAT dated December 15, 2021, with 6.1 million tips, highlighting the newly-emerged BA.1 lineage using a black box. (Right) Expanded view of the BA.1 lineage highlighting the WEPP-detected haplotypes (in circles) for samples between December 16, 2021, and December 29, 2021, with bold borders for haplotypes later confirmed clinically in San Diego at a single-nucleotide substitution distance of 0 or 1 from a WEPP haplotype.

lineage distributions, we performed Kullback-Leibler (KL) divergence analysis, comparing Point Loma wastewater esti-mates to those from "oracle" clinical sequencing (applying a one-week shift to account for earlier detection via wastewa-ter). WEPP's estimates consistently exhibited lower divergence with clinical data (Fig 4C), highlighting improved precision and sensitivity in detecting circulating variants.

Both WEPP and Freyja detected the Omicron variant (B.1.1.529) in wastewater as early as December 1, 2021, using the MAT from the same date. Each tool reported the variant at approximately 2% relative abundance (S4 Table). Notably, WEPP identified two distinct Omicron haplotypes at 0.9% and 1% abundances in that sample. One haplotype was later confirmed by 127 clinical sequences from San Diego beginning December 8, 2021, while the other was observed clinically only three times, starting January 8, 2022. In total, 32 haplotypes identified by WEPP in wastewater were also observed in the clinical data (S5 Table). During the first three weeks of December 2021, when Omicron emerged and rapidly spread in the city, WEPP detected Omicron haplotypes an average of 12.9 days earlier (standard deviation 17.7 days) than clinical sample collection (Fig E in S1 Text). More importantly, WEPP detected multiple distinct haplotype clusters (defined using a radius of two single-nucleotide substitutions) within the B.1.1.529 (Omicron) lineage in wastewater samples as early as December 1, 2, and 5, 2021, revealing three independent Omicron introductions into San Diego (S10 Table and Fig C in S1 Text). By identifying these separate introduction events at low proportions and at least a week before clinical confirmation, WEPP could have guided targeted interventions (e.g., focused testing, enhanced contact tracing) to curb the spread.

Additionally, WEPP identified six intra-lineage haplotype clusters from wastewater within the BA.1 lineage above 1% average abundance, circulating in San Diego between December 16, 2021, and December 29, 2021, as shown in Fig 4D. BA.1 was a newly emerged lineage, and public databases (including GISAID) contained only 11 clinical BA.1 sequence depositions (<0.5% of the total) from San Diego during this period. In contrast, wastewater data showed significant and sharply rising BA.1 prevalence, from about 63% to nearly 95% during the same period, as reported by both WEPP and Freyja (S4 Table). This indicates that local clinical sequencing in San Diego lagged behind and failed to accurately capture the prevailing transmission dynamics. Importantly, five of the six BA.1 clusters identified by WEPP included at least one haplotype that was within a single-nucleotide substitution of a clinical sequence reported from San Diego in the following 2–3 weeks. Additionally, WEPP detected intra-lineage haplotype clusters from each of the 22 wastewater samples analyzed during the entire period (S14 Table). These findings highlight the possibility of multiple independent introductions in the city and underscore WEPP's unique capability to track emerging clusters in a detailed and timely fashion.

We further evaluated the impact of MAT updates on the detection accuracy of both tools by simulating a pandemic setting where UShER MATs are updated slowly, every 15 days. With the MAT update for samples collected post-December 15, 2021, both WEPP and Freyja detected BA.1 at over 60% relative abundance, while prior samples had no estimation of BA.1 due to the absence of BA.1 haplotypes in the older MAT from December 1, 2021 (Fig 4A and S4 Table). Similarly, before January 15, 2022, both WEPP and Freyja detected only the BA.1 lineage (Fig 4A and S4 Table), as the earlier MATs (dated December 15, 2021, and January 1, 2022) did not contain any BA.1.1 haplotypes. However, following the MAT update for samples collected after January 15, 2022, WEPP successfully identified both BA.1 and BA.1.1 lineages in wastewater, whereas Freyja detected only BA.1.1. Retrospective analysis of clinical sequences from San Diego, which were unavailable at the time due to reporting delays [58], confirms that each of BA.1 and BA.1.1 accounted for more than 15% of cases from mid-December onward, with BA.1.1 increasing in prevalence over time—a trend more closely mirrored by WEPP's results (Fig 4B and S4 Table). This underscores WEPP's enhanced sensitivity in resolving closely related lineages from real wastewater data, which would enable epidemiologists to more accurately track pathogens' transmission and evolutionary dynamics.

To assess WEPP's performance under noisy sequencing conditions, we also analyzed high-error, long-read ONT-sequenced wastewater samples from Columbia University's hospital. For both WEPP and Freyja, we averaged the estimated lineage abundances across all samples and computed the KL divergence against lineage abundances from clinical sequences collected during the same month. The KL divergence values were comparable for both tools (Fig A(ii) and A(iii) in S1 Text and S3 Table), demonstrating that WEPP maintains robust performance under noisy sequencing conditions.

Collectively, these results highlight WEPP's robustness across sequencing technologies, demonstrating high sensitivity and accuracy towards emerging variant detection, even at very low abundances, and identifying several intra-lineage haplotype clusters as well as multiple local cluster introductions up to weeks ahead of clinical confirmation. This early signal could provide critical lead time for public health officials to respond and mitigate potential large-scale outbreaks. These

findings underscore that WEPP would enable more effective and timely interventions as well as demonstrate the necessity of frequent MAT updates for sustaining high accuracy in wastewater-based surveillance.

## WEPP generalizes across pathogens and detects lineages missed by clinical surveillance

Wastewater-based surveillance is increasingly being expanded to track a range of infectious diseases beyond SARS-CoV-2, including Influenza A, RSV, Mpox, Measles, and Hepatitis. However, unlike SARS-CoV-2, none of these pathogens are extensively sequenced, which places greater importance on WBE tools that can generalize well and fully leverage all the available sequence data.

To demonstrate WEPP's pathogen-agnostic capabilities, we evaluated its lineage estimation capabilities on two other pathogens: Respiratory Syncytial Virus Subgroup A (RSV-A) and Mpox, using three simulated datasets for each pathogen (Fig 5A; see Methods). WEPP accurately detected the abundances of all the expected lineages in both cases, validating its applicability across diverse pathogens.

We further evaluated WEPP's performance on real RSV-A wastewater samples using data from Korne-Elenbaas et al. [35], collected in Geneva and Zurich during the 2023–2024 RSV-A outbreak. Relative to V-Pipe [20], the method used in the original study, WEPP, consistently detected additional lineages—A.D and A.D.5—at notable proportions across multiple days in both Geneva and Zurich (Fig 5B and Fig 5C and S8 Table). WEPP's enhanced detection is attributed to its comprehensiveness, as it used the entire RSV-A phylogeny, whereas the V-Pipe analysis was limited to eight lineages derived from the 51 RSV-A genomes reported in Switzerland during that period. To further validate these findings, we independently analyzed a subset of samples using Freyja, which also detected the two additional lineages identified by WEPP in both locations, strengthening confidence in WEPP's results.

These results underscore WEPP's high sensitivity and adaptability in detecting circulating lineages of diverse pathogens from wastewater, including those missed by clinical surveillance, by leveraging both observed and inferred ancestral sequences across the entire phylogenetic tree. This approach is particularly valuable for pathogens with sparse clinical sequencing, where existing WBE tools would struggle.

## Discussion

WBE offers a powerful tool to fight local epidemics and outbreaks, and is helping mitigate the huge disparities that exist globally in clinical surveillance capabilities [61]. Although global clinical sequencing and data-sharing efforts for infectious pathogens have greatly accelerated in recent years, particularly since the onset of the COVID-19 pandemic, this massive wealth of clinical sequencing data has largely been underutilized in WBE. For example, despite over 16 million SARS-CoV-2 isolates sequenced globally to date [62], most COVID-19 WBE studies heavily subsample this data, discarding over 99–99.9% of available clinical isolates through aggressive pre-filtering strategies [36] or limiting searches to the roots of just a few thousand named lineages [2,21–24,26].

WEPP overcomes these limitations by placing sequencing reads onto comprehensive mutation-annotated phylogenetic trees (MATs) that include all global clinical sequences and their inferred ancestors. Through a suite of algorithmic and software optimizations (Methods), WEPP efficiently analyzes millions of reads against tens of millions of haplotype candidates, improving WBE in three ways.

1. **Improved accuracy:** WEPP markedly improves lineage abundance estimates compared to existing tools that limit searches to lineage roots [2] or apply aggressive prefiltering to select haplotypes [36].

2. **Enhanced resolution:** By effectively leveraging mutation-annotated trees (MATs)—UShER-based phylogenies that are updated daily, annotated with lineage labels, and include both global clinical sequences and their inferred ancestors—WEPP can sensitively and precisely identify likely phylogenetic haplotypes present in wastewater. For densely-sampled pathogens like SARS-CoV-2, this enables near-haplotype-level resolution, unlocking new applications such as tracking

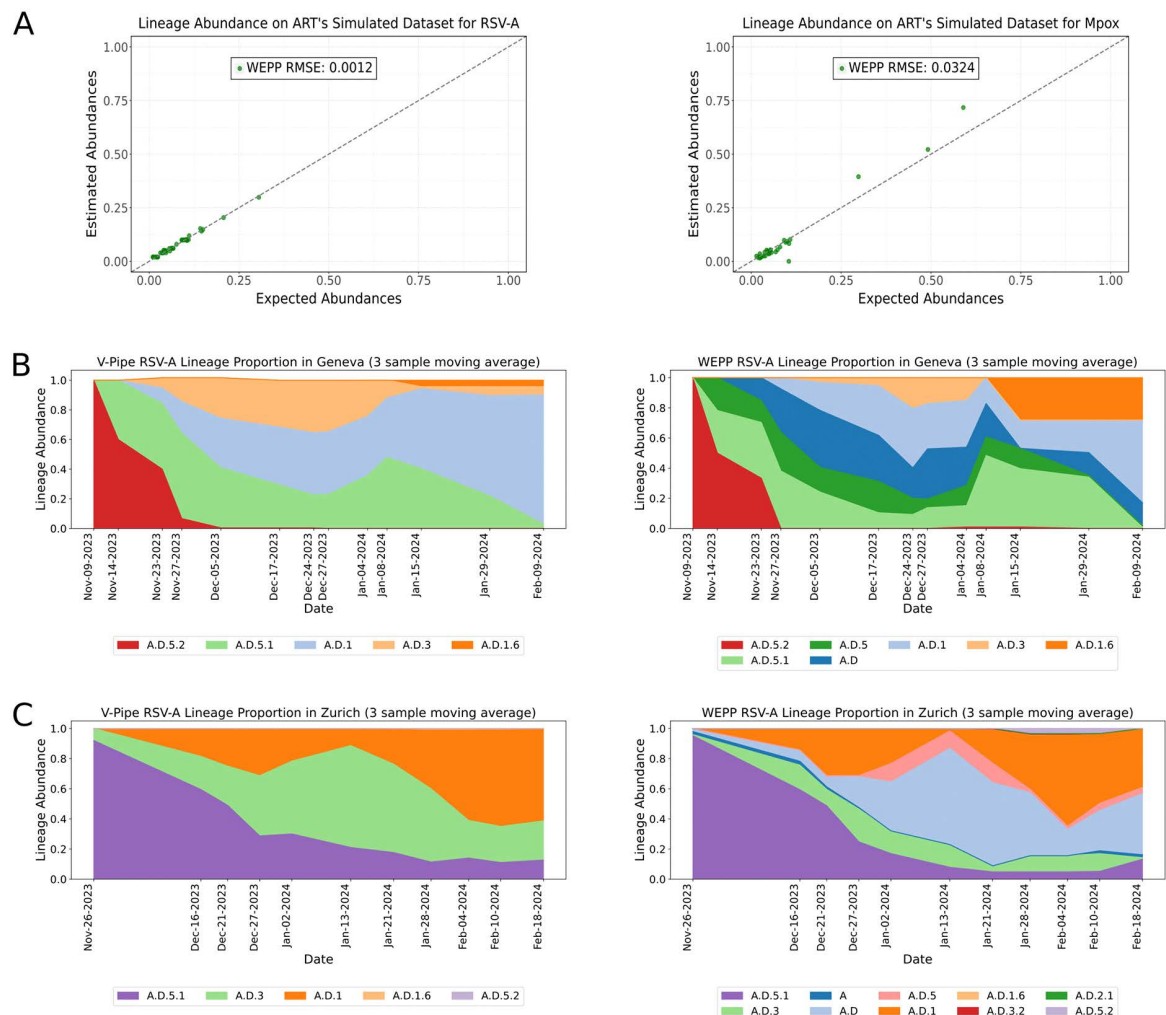

**Fig 5. Analyzing WEPP's lineage abundance results on non-SARS-CoV-2 pathogens. (A)** Left: Panel shows WEPP's lineage abundance accuracy on simulated RSV-A datasets. Right: WEPP's lineage abundance results on simulated Mpox datasets. **(B)** Left: V-Pipe's lineage abundance results on wastewater samples from Geneva, Right: WEPP's abundance estimates on Geneva's wastewater samples. **(C)** Left: V-Pipe's lineage abundance results on wastewater samples from Zurich, Right: WEPP's abundance estimates on Zurich's wastewater samples.

intra-lineage clusters, identifying the geographic origins of newly introduced clusters, and detecting undesignated emerging variants (including recombinants).

3. **Greater timeliness:** Because WEPP selects haplotypes directly from frequently updated MATs, it can bypass delays associated with the formal designation of new lineages, which often lag sequence collection by days to months [63]. This allows WEPP to rapidly detect whether newly identified clinical variant clusters from elsewhere in the world are present in local wastewater samples. Moreover, WEPP's reporting of unaccounted alleles enables the timely identification and analysis of cryptic or novel variants.

In terms of limitations, while WEPP enables new capabilities for WBE, it relies on comprehensive global clinical sequencing data organized in the form of MAT. However, these MATs are also required by some other WBE tools, such as Freyja, and are now maintained for many pathogens (see dev.usher.bio). Moreover, the recently introduced viral_usher

(https://github.com/AngieHinrichs/viral_usher) automates the MAT creation for any virus. As sequencing costs continue to decline and an increasing number of pathogens are being densely sequenced, and with ongoing efforts to extend viral_usher to support bacterial species, WEPP's utility is expected to expand to a broader range of pathogens. Even when pathogens are less densely sampled (like RSV and Mpox currently), WEPP remains effective because of its ability to flag new or previously unseen mutations as 'unaccounted alleles'. Another limitation is that WEPP requires sufficiently high coverage and sequencing quality to provide accurate and confident results. Empirically, for short-read sequencing, we have found that WEPP achieves the most accurate results when the sequencing data has a median Phred score of 38 or above and a per-haplotype coverage of above 25× (Fig F in S1 Text). WEPP's analysis, while independently useful, can also complement several existing WBE tools. For example, WEPP can be integrated into multi-pathogen wastewater sequencing pipelines, such as the one developed by Tisza et al. [64], to enhance the resolution of metagenomic surveillance and enable early detection of emerging variants. The unaccounted alleles identified by WEPP can also be integrated with mutation fitness estimators [65], as well as multi-sample frameworks that include clustering and reproduction number estimation [26] and spatiotemporal analysis [66]. This integration could enable a more comprehensive assessment of novel variants from multiple wastewater samples. Likewise, while WEPP reports uncertainty in haplotype detection based on the informative sites available in wastewater, its abundance estimates can be combined with methods like Dreifuss et al. [67] to assess uncertainty in abundance quantification. Additionally, WEPP's dashboard provides fine-grained visualizations of read mappings, unaccounted alleles, and haplotype selections within the context of global clinical data, complementing other wastewater dashboards [68–71] that focus on broader lineage-level or viral concentration summaries. Our future efforts would involve integrating WEPP with these complementary tools. We also plan to incorporate both insertions and deletions into WEPP's analysis with the help of recently developed PanMATs [72], although our preliminary analysis suggests that incorporation of deletions does not improve the accuracy of lineage abundances and haplotype detection (Supplementary Results 1 and Fig D in S1 Text). We will also explore further algorithmic and acceleration techniques, such as phylogenetic placement heuristics [73] and GPU acceleration [74,75], to further enhance the scale and efficiency of WEPP.

As a summary, WEPP's integration of clinically derived global phylogenetic data with wastewater surveillance enables high-resolution analysis and marks a transformative advance in pathogen monitoring. Importantly, WEPP maintains robust performance across multiple pathogens as well as on error-prone long-read data (e.g., Oxford Nanopore Technologies), achieving a 3.2-fold lower RMSE than Freyja (Supplementary Results 2 in S1 Text). This positions WEPP to significantly strengthen and expand the scope of current wastewater surveillance efforts, enabling faster and more precise analysis of emerging threats posed by various pathogens, especially in the post-pandemic era.

## Methods

### Datasets

**Mutation-Annotated Trees (MATs) and Pangenome Mutation-Annotated Trees (PanMATs).** The MAT used for the analysis of simulated wastewater datasets was dated December 25, 2023, and comprised publicly available sequences aggregated from GenBank, COG-UK, and the China National Center for Bioinformation (https://hgdownload.gi.ucsc.edu/goldenPath/wuhCor1/UShER_SARS-CoV-2). The tree was rooted to the GenBank reference sequence MN908947.3 and consisted of 7,840,184 sequences. In contrast, analyses of the synthetic wastewater datasets, as well as samples from Annavajhala et al. [10] and Suarez et al. [37], were performed using the MAT dated December 15, 2023, which additionally incorporated restricted sequences from the Global Initiative on Sharing All Influenza Data (GISAID), resulting in a total of 15,935,784 sequences. Analyses of RSV-A and Mpox datasets were conducted using publicly available sequences using the corresponding MATs dated April 25, 2025 (https://genome-test.gi.ucsc.edu/cgi-bin/hgPhyloPlace). All SARS-CoV-2 MATs were pre-annotated with PANGO [76] and Nextclade [77] lineage annotations, while Mpox and RSV-A MATs contained only Nextclade annotations.

PanMAT is a mutation-annotated tree that also stores insertions and deletions along with substitutions and uncertain alleles [72]. We constructed a PanMAT incorporating substitutions, deletions, and uncertain alleles using sequences from the MAT dated December 25, 2023, after excluding 45,144 haplotypes lacking corresponding FASTA files, resulting in a total of 7,840,184 sequences. Each sequence was pairwise aligned to the reference genome (GenBank: MN908947.3) using the Nextclade aligner [77] that excluded insertions. The resulting MSA and MAT-derived newick tree was used to generate the PanMAT with the command: "*./panmanUtils --input-msa <file.fa> --input-newick <file.nwk> -o <file.panman> --low-mem-mode --reference Wuhan-Hu-1*". The PanMAT was used to evaluate the impact of incorporating deletions into WEPP's analysis (Supplementary Results 1 in S1 Text).

### Wastewater datasets

To comprehensively evaluate WEPP's performance, we used a combination of simulated datasets and synthetic control mixtures with known ground truths, along with real wastewater samples. Below, we detail the construction and characteristics of each dataset.

- **Simulated wastewater datasets:** We generated in silico SARS-CoV-2 wastewater datasets using the SWAMPy simulator [47], guided by lineage proportions observed at Point Loma between December 2022 and December 2023. Each simulated sample contained 100 haplotypes drawn proportionally from the observed lineage distributions. SWAMPy was used with the following parameters to generate 1.6 million reads: "*--primer_set n2 --n_reads 800000 --read_length 149*", while "*--minQ 25*" was set in the "art_runner.py" script to reflect typical wastewater read quality profiles.

  For the samples used to mimic realistic cluster emergence scenarios, we first removed the haplotypes newly emerged (via ClusterTracker [31]) in the target region (e.g., haplotypes emerging in California, during New York to California transmission) from the MAT and simulated 5% of wastewater reads from one of the removed haplotypes, while the remaining 95% were drawn from lineages circulating locally in the target region at that time.

  For the experiments involving the analysis of unaccounted alleles, we used ART simulator [78] instead of SWAMPy to ensure that the novel mutations were not dropped by the simulator. For these experiments, we created four base mixtures and simulated each with a novel haplotype at 5%, 10%, or 20% abundance, resulting in 12 samples with 1.6 million reads each. The unique haplotypes in these mixtures ranged from 78 to 96, the number of alleles from 762 to 1,018, and the average minimum pairwise distance between haplotypes from 4.64 to 6.97 per sample (S13 Table). These were evaluated on phylogenetic trees where the novel haplotype and its surrounding nodes within 3, 5, or 7 mutation distances were removed, producing a total of 36 distinct test cases.

  To evaluate WEPP's ability to detect recombinants, we used ART to simulate two samples containing a Delta-Omicron recombinant, XAY, and one Omicron-Omicron recombinant, XBF. To make the analysis more challenging, we selected two haplotypes from their parental lineages for each sample and simulated the recombinant and two parental haplotypes at equal proportions. We evaluated the scenario of detecting *known* recombinants, i.e., when they are present in the MAT, as well as *novel* recombinants, i.e., when the recombinant lineage is absent in the MAT. For the latter, the recombinant and all of its descendant lineages were removed from the input MAT.

  As SWAMPy only supports SARS-CoV-2, ART was also employed to simulate wastewater datasets for RSV-A and Mpox. For these pathogens, we generated samples containing 10, 20, or 50 randomly selected haplotypes.

- **Synthetic control datasets**: To compare WEPP and Freyja's performance on synthetic controlled mixtures, assay-ready RNA controls from Twist Biosciences (https://www.twistbioscience.com/products/ngs/synthetic-viral-controls?tab=sars-cov-2-controls) were spiked into SARS–CoV–2–negative wastewater to generate synthetic samples. We benchmarked WEPP on both Illumina-sequenced mixtures prepared by the Indiana Department of Health (IDOH) and Oxford Nanopore Technologies (ONT)-sequenced mixtures from Ferdous et al. [46], to assess generalizability across sequencing technologies.

The IDOH samples were produced by creating five different synthetic mixtures; three of which contained three haplotypes in equal proportions, while the remaining two contained a single haplotype. Each sample was independently prepared and sequenced twice to assess wet-lab reproducibility, yielding a total of 10 samples.

Additionally, WEPP was tested on three ONT-sequenced synthetic mixtures from using R9.4.1 flow cells, Ferdous et al., each comprising eight distinct haplotypes in varying proportions. Individual haplotype abundances ranged from as low as 6.25% to as high as 31.25%, allowing us to assess WEPP's performance under challenging abundance distributions and sequencing noise.

- **Real wastewater samples:** To evaluate WEPP's performance on real-world data, we analyzed 77 wastewater samples from four different sources to assess a range of surveillance goals: correlation with clinical sequencing, early detection of novel SARS-CoV-2 variants and mutations, cryptic lineage detection, and generalizability to non-SARS-CoV-2 pathogens. For most real datasets below involving short-read sequencing, the median Phred score ranged from 37 to 39, which is similar to our simulations.

  1. **SARS-CoV-2 wastewater from Point Loma, San Diego (22 samples)**

     We analyzed 22 samples collected between November 27, 2021, and February 7, 2022, from San Diego's largest wastewater treatment plant, as described in Karthikeyan et al. [2] (BioProject Accession ID: PRJNA819090). To mimic real-time surveillance, each sample was analyzed using MATs that also included private sequences from GISAID, which were constructed before the sample's collection date. Specifically, samples collected before December 1, 2021, used the November 15, 2021, MAT; subsequent samples used MATs that were updated every 15 days. WEPP's lineage abundance estimates were compared with those from Freyja and "oracle" clinical sequencing data. Unaccounted alleles identified by WEPP were also compared against all sequences reported from San Diego from November 2021 to March 2022, to assess their rarity and potential for cryptic mutations. We also analyzed the WEPP-inferred haplotypes and compared them against all clinical sequences from San Diego collected during the same period, along with their earliest detection dates, to evaluate how accurately and early WEPP could detect these sequences from wastewater.

  2. **SARS-CoV-2 samples from NewYork-Presbyterian Hospital, Columbia University Irving Medical Center (13 samples)**

     We analyzed 13 wastewater samples collected in December 2022 from the New York-Presbyterian Hospital at Columbia University Irving Medical Center, from Annavajhala et al. [10] (BioProject Accession ID: PRJNA1230707), which were sequenced with ONT using R9.4.1 flow cells. WEPP's and Freyja's lineage abundance estimates were compared against 57 clinical sequences obtained from the same hospital and time.

  3. **Cryptic lineage containing SARS-CoV-2 samples from Suarez et al. (18 samples)**

     We evaluated 18 wastewater samples containing different cryptic SARS-CoV-2 lineages as described in Suarez et al. [37]. For each sample, we focused on "cryptic lineage–defining amino acid substitutions" as reported in the original study. These were translated into nucleotide-level mutations from their data, and in case of multiple codons encoding the same amino acid change, we selected the nucleotide mutation least frequent in WEPP's inferred haplotypes. We then compared these mutations against WEPP's unaccounted allele calls, while also analyzing them across all WEPP's thresholds, namely coverage depth and deviation in depth-weighted allele frequency from expectation.

  4. **RSV-A samples from Geneva and Zurich (24 samples)**

     To test WEPP's generalizability, we analyzed 24 RSV-A samples collected during the 2023–2024 outbreak from wastewater in Geneva and Zurich, as shared by De Korne-Elenbaas et al. [35] (ENA Study Accession: PRJEB85787). WEPP's lineage abundance estimates were then compared to those reported by V-Pipe, which was restricted to lineages previously observed in Swiss clinical sequences during the outbreak period [35].

## Baseline tools

We compared WEPP's performance against two baselines: Freyja [2] and the EM algorithm proposed by Pipes et al. [36]. Freyja has consistently been ranked among the most accurate WBE tools in multiple benchmarking studies [46,79] and is also widely used in practice [10,13,50,80,81]. The EM algorithm proposed by Pipes et al. remains the only other available method for estimating haplotype abundances from wastewater data.

   We integrated Freyja into WEPP and, when benchmarking against it, used lineage roots as input in place of phylogenetically selected haplotypes (see our implementation at https://github.com/TurakhiaLab/WEPP/tree/wepp_freyja). For the EM baseline, we refactored and reimplemented the algorithm from Pipes et al. in C++ to improve computational efficiency (see our implementation at https://github.com/TurakhiaLab/WEPP/tree/wepp_em), and applied a default haplotype filtering threshold of 1% allele frequency and sequencing error rate of 0.5%, as described in their paper. Inputs to the EM algorithm were generated by computing mutation differences between sequencing reads and MAT sequences. The algorithm was terminated when the log-likelihood change between successive iterations dropped below 0.1% or after a maximum of 25 iterations.

## WEPP: Pipeline overview

The WEPP pipeline is a Snakemake workflow that consists of three main components: (i) Quality Control filtering and Read Subsampling, (ii) WEPP Algorithm, and (iii) WEPP Dashboard analysis. The pipeline requires as input: (i) raw wastewater sequencing reads, (ii) mutation-annotated tree, (iii) corresponding reference genome, and (iv) BED file specifying primer positions.

## Quality control filtering and read subsampling

The quality control begins by aligning raw sequencing reads to the reference genome using minimap2 [82]. Aligned reads are then processed with iVar [83] for primer trimming based on the provided BED file. WEPP then applies two layers of allele-level quality control to ensure reliability in downstream analysis. First, bases with a Phred quality score below a user-defined threshold (MIN_Q, default: 20) are masked within the reads. Second, alleles with observed frequencies below a specified minimum allele frequency (MIN_AF, default: 0.5% for Illumina reads) are also masked in the reads. Importantly, all reads are retained in the dataset, but their sites failing these QC filters are masked and excluded from downstream analysis.

   WEPP includes an optional read subsampling feature to reduce the runtime. It generates $N$ independent subsets (default: 1000 subsets), each containing a user-specified number of randomly selected reads and selects the subset whose allele frequency distribution has the lowest Kullback-Leibler (KL) divergence from the full dataset, ensuring minimal distortion of information.

## WEPP: Algorithm details

WEPP consists of three main processing stages: (i) Read Mapping and Haplotype Selection, (ii) Neighbor Addition, and (iii) Abundance Estimation (Fig 1C). Once WEPP infers the haplotypes, unaccounted alleles are computed, and reads are mapped to these inferred haplotypes. The core algorithm is implemented as multi-threaded C++ using the oneTBB library [84], which is integrated with the Quality Control pipeline and interactive WEPP Dashboard via a Snakemake workflow. Detailed implementation of different stages is described below.

1. **Read Mapping and Haplotype Selection:** WEPP processes all sequencing reads after the quality control stage and identifies genome sites that remain uncovered. It performs haplotype grouping by clustering haplotypes that differ only at these uncovered positions and reports them as 'Uncertain Haplotypes' for each selected representative haplotype (Fig 1B(i)). For each read, WEPP computes two values: the parsimony [85] (i.e., minimum number of mutations required to place a read on the phylogenetic tree) and the number of equally parsimonious placements (EPPs). Both of these values are then used to assign a weight to every read.

Once all reads are mapped to the tree, WEPP assigns a score to each node by summing the weights of all reads that map parsimoniously to it. This score is then regularized by the fraction of the genome covered by those reads.

Next, WEPP applies a greedy haplotype selection algorithm. It begins by selecting the node with the highest score as the initial haplotype. After the haplotype is selected, the scores of all remaining haplotypes are updated by subtracting the weights of reads that map parsimoniously to the selected haplotype. This process of haplotype selection followed by score update is repeated iteratively until all reads are either exhausted or a predefined number of haplotypes are selected (default: 300).

$$Read\ Weight\ (W_i)\ =\ \left((1\ +\ Parsimony)\ *\ EPP\right)^{-1}$$

$$Node\ Score\ (N_j)\ =\ \left(\sum_{i=1}^{m} W_i\right)\ *\ \left(Coverage\ Fraction\right)^{1/2}$$

2. **Neighbor Addition:** After identifying an initial set of haplotypes using the greedy haplotype selection algorithm, WEPP expands this candidate set by including neighboring nodes around each haplotype. The mutation radius used to define these neighbors is chosen adaptively to maintain the total size of the final candidate set at a fixed level (default: 5,000 haplotypes). To further constrain the search space, no more than a fixed number of neighbors (default: 50) are added per initially selected haplotypes. Neighbor addition ensures that closely related haplotypes, which differ by only a few mutations, are not overlooked.

3. **Abundance Estimation:** WEPP applies an extended version of Freyja's abundance estimation framework to the expanded haplotype set generated after Neighbor Addition. To reduce overfitting and noise from spurious matches, haplotypes with low abundance estimates (default: 0.5%) are greedily pruned. Starting with the lowest-abundance haplotype, each haplotype below the threshold is removed and its proportion reassigned to the closest neighbor from the candidate set. Once the proportion of all remaining haplotypes is above the threshold, Freyja is rerun to refine the estimates, and any newly sub-threshold haplotypes are discarded. This greedy approach avoids the pitfall of naively discarding all haplotypes below the abundance threshold, where an entire haplotype cluster gets eliminated if all its members fall below the threshold, despite collectively representing a significant abundance. By reassigning proportions before removal, WEPP preserves the cluster's representation and improves robustness.

   As illustrated in Fig 1C, WEPP then enters an iterative refinement loop, where neighbors of the retained haplotypes are re-added to the candidate pool (default: mutation distance radius of 2; maximum 500 neighbors per haplotype), and the abundance estimation step is repeated. This process continues until the retained haplotype set converges (i.e., remains unchanged over two successive iterations) or a maximum number of iterations (default: 10) is reached.

4. **Unaccounted Alleles:** WEPP extends Freyja's optimization framework to identify *unaccounted alleles*, alleles present in wastewater but not explained by the selected haplotypes. While Freyja minimizes the depth-weighted absolute differences between observed and predicted allele frequencies (*weighted freq_diff*), WEPP builds on it by first computing the mean and standard deviation of these differences across all mutations. Mutations that significantly deviate from this distribution, i.e., two standard deviations from the mean value, are flagged as unaccounted, with their direction of deviation indicating whether the allele is over- or under-represented in the selected haplotypes. A mutation is labeled as an unaccounted allele if it satisfies all the following criteria:

   a. *weighted freq_diff$_i$* $\notin$ [*mean* $-2 *$ *std deviation*, *mean* $+2 *$ *std deviation*]

   b. *Depth$_i$* $>$ $0.6 *$ *mean Depth*

   c. |*freq_diff$_i$*| /*Observed_freq$_i$* $>$ $0.5$

5. **Read-to-Haplotype Mapping and Parsimonious Haplotype Candidates for unaccounted alleles:** After identifying all haplotypes present in a sample, WEPP performs parsimony-based mapping of each read to this selected set of haplotypes to determine its most likely origin. To associate each unaccounted allele with a parsimonious haplotype candidate, WEPP first masks the unaccounted alleles in the reads and groups the reads based on the specific unaccounted allele they contain. Within each group, it selects the haplotype that receives the highest number of parsimonious read mappings and records it as the most likely source of that unaccounted allele.

## WEPP: Optimization details

Mapping more than 4 million sequencing reads from wastewater to a phylogenetic tree of ~16 million SARS-CoV-2 sequences is computationally challenging. To make this process feasible in near real-time, WEPP employs several key optimizations:

1. **De-duplication of reads.** WEPP first identifies and collapses duplicate reads, performing parsimony-based mapping only for this unique set. Node scores are then adjusted by the frequency of each unique read in the dataset, greatly reducing redundant computations.

2. **Parallelized read mapping:** WEPP extracts parallelism during the read mapping stage by assigning each read to a separate thread, allowing multiple reads to be processed simultaneously. This thread-level parallelism significantly accelerates the search and improves scalability in the presence of a large number of unique reads.

3. **Phylogenetic tree condensation.** To minimize unnecessary traversal and track 'Uncertain Haplotypes' (Fig 1B(i)), WEPP prunes the global phylogenetic tree by retaining only those nodes with mutations at positions covered by the wastewater reads. It stores the condensed node information at each retained node, resulting in a much smaller and more relevant subtree that significantly accelerates the mapping process.

4. **Region-aware tree search.** WEPP partitions the reads based on the genome segment they cover and constructs localized subtrees that only contain mutations in those regions. This region-aware approach avoids checking unrelated nodes with no relevant mutations, leading to faster and more efficient parsimony computation.

## Evaluation metrics

We evaluated the performance of WEPP against baseline methods using simulated and synthetic control mixtures. Comparisons were based on three key metrics: Lineage Abundance RMSE, Weighted Haplotype Distance, and Weighted Peak Distance.

- **Lineage Abundance RMSE**: Measures the root mean squared error between the expected and estimated lineage abundances.

- **Weighted Haplotype Distance**: Assesses how well the most abundant haplotypes in the sample are recovered by each tool, with greater weight given to haplotypes of higher proportion. Each true haplotype is matched to the closest estimated haplotype based on minimum mutation differences. This metric reflects *sensitivity*; lower values indicate better recovery of true haplotypes.

- **Weighted Peak Distance**: Captures *precision* by evaluating how closely the highest estimated abundance haplotypes reported by a tool match true haplotypes present in the wastewater sample. Lower values of this metric indicate fewer spurious inferred haplotypes.

The weighted distance metrics can be defined more formally by assuming H={$h_1,h_2,\ldots,h_n$} as the set of true haplotypes in the sample with corresponding proportions wh, and P={$p_1,p_2,\ldots,p_m$} as the set of inferred haplotypes with corresponding

proportions wp. The mutation distance captures the number of single-nucleotide substitutions separating the true haplotype $h_i$ from the inferred haplotype $p_j$, which is denoted by $Dist(h_i, p_j)$.

$$Weighted\ Haplotype\ Distance\ =\ \sum_{i=1}^{n} wh_i * min_{\forall j}\ Dist(h_i,\ p_j)$$

$$Weighted\ Peak\ Distance\ =\ \sum_{j=1}^{m} wp_j^* min_{\forall i}\ Dist(p_j,\ h_i)$$

### Execution environment and runtimes

All experiments were performed on an Intel Xeon Silver 4216 processor with 768GB DDR4 memory and 4.0TB SATA. WEPP was run using 48 CPU threads, with runtimes varying from a few minutes to 14 hours depending on the size of the MAT and the number of reads in the sample (Fig G in S1 Text). For example, the RSV-A datasets from Korne-Elenbaas et al. [35] contained approximately 2 million sequencing reads, which were analyzed in under 10 minutes, since the RSV-A MAT contained only 8,215 sequences. In contrast, Point Loma wastewater samples were analyzed in approximately 4 hours, containing a similar number of reads due to a much larger SARS-CoV-2 MAT comprising 7.45 million publicly available sequences. The longest runtimes were observed for synthetic samples from IDOH, which were analyzed using an even larger MAT that also included restricted GISAID sequences, increasing its size by 2.14-fold (15.93 million sequences). Additionally, these IDOH samples contained about 25% more unique reads than the Point Loma samples, even after subsampling to 1 million reads, which limited the effectiveness of certain WEPP optimizations (see above).

### WEPP dashboard: Overview and implementation details

To enhance the utility and interpretability of WEPP results, we developed an interactive visualization dashboard using the Taxonium framework [45] and a custom JBrowse plugin [86], which enables researchers and public health officials to interpret wastewater surveillance results more effectively. The dashboard displays detected haplotypes, their corresponding lineages, along with associated abundances and mutation profiles in an intuitive visual format. This would enable public health officials to rapidly identify emerging haplotype clusters and lineages, monitor shifts in their prevalence over time, and make informed, data-driven decisions for targeted community health interventions (Fig 1B).

The dashboard customizes the Taxonium framework [45] for visualizing detected haplotypes within their evolutionary context. This interactive component enables users to navigate the complete phylogenetic tree through intuitive pan and zoom controls while accessing contextual information by hovering over specific nodes. The dashboard's search panel displays detected haplotypes with their corresponding abundances, lineage classifications, and associated uncertain haplotypes, enabling rapid identification of variants of interest (Fig 1B(i)).

When users select an annotated node on the dashboard, a custom JBrowse plugin [86] is triggered to enable high-resolution sequence analysis. This plugin provides an interactive, multi-track genome browser view that includes: (i) annotated gene features along the reference genome, (ii) reconstructed haplotype sequences aligned to the reference, and (iii) individual sequencing reads from the wastewater sample, mapped to both the reference and haplotype sequences. This layered visualization enables users to explore how specific mutations are supported by sequencing reads and to assess their linkage within the context of gene structure and inferred haplotypes (Fig 1B(ii)). Additionally, users can click on any individual sequencing read within the BAM track to explore alternative haplotype associations and view unaccounted alleles (Fig 1B(iii)). Similarly, clicking on a haplotype sequence reveals a list of all possible unaccounted alleles linked to that haplotype, along with the associated uncertain haplotypes (Fig 1B(iv)), which aid in the identification of low-frequency or ambiguous variant signals.

The WEPP dashboard allows users to investigate the uncertainty in the analysis. Specifically, it displays the uncertainty in each haplotype selection by reporting 'Uncertain Haplotypes' (Fig 1B(i)). To distinguish whether this uncertainty arises from identical haplotypes in the MAT or from incomplete genome coverage by sequencing reads, the WEPP dashboard reports 'Maximum Nucleotide Distance' among the uncertain haplotypes. This value is zero in the case of identical haplotypes. Non-zero values imply that haplotypes are non-identical, but there is uncertainty potentially caused by the lack of coverage of informative sites in the sample. The dashboard also reports the percentage of each haplotype's genome covered by parsimoniously mapped reads. In addition, WEPP characterizes uncertainty in read-to-haplotype mappings, which is particularly important for short sequencing reads that contain limited informative sites and therefore map parsimoniously to multiple haplotypes. This uncertainty is quantified using the Equally Parsimonious Positions (EPP), which reports the number of haplotypes to which a read can map with equal parsimony (lower value indicates higher confidence).

To visualize results on this dashboard, the WEPP's pipeline generates two annotated BAM files. The first contains WEPP-inferred haplotypes, with metadata added to the BAM header using read groups (@RG) to identify each unique haplotype. These read groups include tags such as HS (haplotype proportion), HL (lineage label), and UH (uncertain haplotype flag). The second BAM file contains the original wastewater sample reads, annotated with tags like UM to flag unaccounted alleles, RG to link a read to its possible source haplotypes, and EP to indicate the number of parsimonious haplotype candidates. Both BAM files also include @CO comment lines, each of which describes an uncertain allele with a unique identifier, its position and base, associated residue, frequency, and depth.

## Supporting information

**S1 Table. Comparison of lineage abundance estimates of different tools with expected abundances on simulated and synthetic wastewater datasets.**
(XLSX)

**S2 Table. Comparison of lineage abundance estimates of Freyja and WEPP with expected abundances on SWAMPy-simulated wastewater samples, generated with SARS-CoV-2 sequences detected in San Diego during September and January 2023, which were identified by the keyword "CA-SEARCH" from the MAT.**
(XLSX)

**S3 Table. Lineage abundance estimates generated by Freyja and WEPP from wastewater samples collected in December 2022 from Hospital Quadrants A and B, along with the lineage abundances computed from the clinical sequencing data obtained from the same hospital and time period.**
(XLSX)

**S4 Table. Comparison of lineage abundance estimates from Freyja and WEPP on Point Loma (San Diego) wastewater samples collected between November 27, 2021, and February 7, 2022, with one week of delayed clinical sequencing data collected from the city to account for earlier detection from wastewater.**
(XLSX)

**S5 Table. Correlation of WEPP-detected haplotypes with clinical sequences from San Diego labeled 'CA-SEARCH' in the April 15, 2022, comprehensive MAT, considering sequences spanning November–December 2021 and January–March 2022.**
(XLSX)

**S6 Table. Analysis of unaccounted alleles detected by WEPP from Point Loma wastewater samples with respect to the clinical sequences collected from San Diego.** Clinical sequences were obtained from the April 15, 2022,

comprehensive MAT by selecting all 'CA-SEARCH'–labeled sequences spanning November–December 2021 and January–March 2022.
(XLSX)

**S7 Table. Comparing WEPP's lineage abundance estimates with expected abundances on simulated RSV-A and Mpox datasets.**
(XLSX)

**S8 Table. Lineage abundance estimates generated by V-Pipe and WEPP on RSV-A wastewater samples from Geneva and Zurich.**
(XLSX)

**S9 Table. Concordance between WEPP-identified unaccounted alleles and cryptic lineage-defining amino acid substitutions reported by Suarez et al. The table also indicates how many of the cryptic lineage-defining amino acid substitutions satisfy WEPP's depth and allele-frequency thresholds for unaccounted-allele calls.**
(XLSX)

**S10 Table. Omicron (B.1.1.529).** Haplotypes proportions estimated by WEPP from Point Loma wastewater data from Dec 1, Dec 2, and Dec 5, 2021.
(XLSX)

**S11 Table. Comparison of sensitivity and precision of different tools in detecting lineages for simulated and synthetic wastewater datasets.** Only lineages above 0.5% abundance were considered.
(XLSX)

**S12 Table. Evaluating WEPP's recombinant detection capabilities in the presence and absence of recombinants in the MAT.**
(XLSX)

**S13 Table. Wastewater mixture characteristics and WEPP's lineage estimates for samples containing novel haplotypes.**
(XLSX)

**S14 Table. Lineages exhibiting multiple intra-lineage haplotype clusters and unaccounted alleles absent from San Diego clinical sequences, reported from Point Loma (San Diego) wastewater samples collected between November 27, 2021, and February 7, 2022.**
(XLSX)

**S1 Text. The supporting information file contains three supplementary results sections.** Supplementary Results 1: Incorporating deletions via PanMAT in WEPP has a negligible impact on SARS-CoV-2 variant detection accuracy. Supplementary Results 2: WEPP generalizes to noisy long-read sequencing dataSupplementary Results 3: WEPP accurately distinguishes between recombinants and their parental genomes. In addition, the file includes the following figures: Fig A: Comparison of lineage abundance between WEPP and Freyja on ONT sequenced data.Fig B: Single-nucleotide substitutions separating two haplotypes in a lineage. Fig C: B.1.1.529 (Omicron) Haplotypes detected by WEPP from Point Loma (San Diego) wastewater samples dated December 1, 2, and 5, 2021. Fig D Impact of incorporating deletions in WEPP. Fig E Comparison of the earliest wastewater detection of Omicron haplotypes during the first three weeks of December 2021 with their corresponding earliest clinical collection dates in San Diego. Clinical confirmation was established when the corresponding clinical haplotype was within 1 single-nucleotide substitution of the WEPP-identified haplotype. Fig F Weighted

Haplotype Distance and Weighted Peak Distance as a function of (A) sequencing depth, and (B) Phred quality score. Fig G: WEPP's runtime as a function of sequencing read count and the number of sequences in the MAT.
(DOCX)

## Acknowledgments

We gratefully acknowledge all data contributors—namely, the authors and their originating laboratories responsible for specimen collection, and the submitting laboratories for generating and sharing sequence data and metadata via GISAID, GenBank, COG-UK, and the China National Center for Bioinformation—on which this research is based. We thank David Schaeper and the Indian Department of Health for providing the synthetic wastewater datasets. We are also grateful to the authors of Ferdous et al. [46], particularly Samuel Kunkleman, for assistance in accessing their benchmarking data. We thank Medini Annavajhala for sharing both wastewater and clinical samples from her study. We are deeply appreciative of Angie Hinrichs for providing access to private MATs containing GISAID sequences, along with her technical guidance and insightful feedback. We also thank Kristian Andersen, Joshua Levy, Karthik Gangavarapu, Russell Corbett-Detig, Jason Caravas, and Daniel Cornforth for their valuable feedback. We acknowledge Sumit Walia for his contributions in generating the PanMAT and helping it integrate into WEPP.

## Author contributions

**Conceptualization:** Pranav Gangwar, Yatish Turakhia.

**Data curation:** Pranav Gangwar, Yatish Turakhia.

**Formal analysis:** Pranav Gangwar, Pratik Katte, Manu Bhatt, Yatish Turakhia.

**Funding acquisition:** Yatish Turakhia.

**Investigation:** Pranav Gangwar, Manu Bhatt, Yatish Turakhia.

**Methodology:** Pranav Gangwar, Pratik Katte, Manu Bhatt, Yatish Turakhia.

**Project administration:** Yatish Turakhia.

**Resources:** Yatish Turakhia.

**Software:** Pranav Gangwar, Pratik Katte, Manu Bhatt.

**Supervision:** Yatish Turakhia.

**Validation:** Pranav Gangwar, Pratik Katte, Manu Bhatt.

**Visualization:** Pranav Gangwar, Pratik Katte.

**Writing – original draft:** Pranav Gangwar, Pratik Katte, Yatish Turakhia.

**Writing – review & editing:** Pranav Gangwar, Pratik Katte, Manu Bhatt, Yatish Turakhia.

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
