## [Decision Letter · Decision Letter 0]

28 Oct 2025

PCOMPBIOL-D-25-01603

WEPP: Phylogenetic Placement Achieves Near-Haplotype Resolution in Wastewater-Based Epidemiology

PLOS Computational Biology

Dear Dr. Turakhia,

Thank you for submitting your manuscript to PLOS Computational Biology. After careful consideration, we feel that it has merit but does not fully meet PLOS Computational Biology's publication criteria as it currently stands. Therefore, we invite you to submit a revised version of the manuscript that addresses the points raised during the review process.

Please submit your revised manuscript within 60 days Dec 28 2025 11:59PM. If you will need more time than this to complete your revisions, please reply to this message or contact the journal office at ploscompbiol@plos.org. Please include the following items when submitting your revised manuscript:

We look forward to receiving your revised manuscript.

Kind regards,

Lin Hou

Academic Editor

PLOS Computational Biology

Thomas Leitner

Section Editor

PLOS Computational Biology

**Journal Requirements:**

At this stage, the following Authors/Authors require contributions: Pranav Gangwar, Pratik Katte, Manu Bhatt, and Yatish Turakhia. Please ensure that the full contributions of each author are acknowledged in the "Add/Edit/Remove Authors" section of our submission form.

Potential Copyright Issues:

i) Figure 1. Please confirm whether you drew the images / clip-art within the figure panels by hand. If you did not draw the images, please provide (a) a link to the source of the images or icons and their license / terms of use; or (b) written permission from the copyright holder to publish the images or icons under our CC BY 4.0 license. Alternatively, you may replace the images with open source alternatives. See these open source resources you may use to replace images / clip-art:

ii) The following Figure contains screenshots: 1. We are not permitted to publish these under our CC-BY 4.0 license, websites are usually intellectual property and are copyrighted.This includes peripheral graphics of the web browser such as icons and button. We ask that you please remove or replace it.

2) State what role the funders took in the study. If the funders had no role in your study, please state: "The funders had no role in study design, data collection and analysis, decision to publish, or preparation of the manuscript.".

**Reviewers' comments:**

Reviewer's Responses to Questions

Reviewer #1: "WEPP: Phylogenetic Placement Achieves Near-Haplotype Resolution in Wastewater-Based Epidemiology" by Pranav Gangwar et al. is an ambitious and timely approach to augment current possibilities of genomic WBE. I think that this contribution does not only serve as a rich inspiration of further research, but is highly relevant to practitioner from the applied public health side. Hence, I would like to strongly support the publication of the manuscript.

Having said this, from my point of view, beside minor issues listed below, the manuscript in its current form suffers from two, probably related issues, I would like to see addressed, prior to publication. (i) I found the use of the terms "lineage", "haplotype", ("genotype",) "allele" confusing. A precise introduction of the terminology in the context of amplicon sequencing of mixed samples would very much improve the readability. (ii) In this context, the term "Potential Haplotypes for unaccounted alleles" is particular difficult. Related to the latter, I also have to admit that I could not fully follow how the "Read-to-Haplotype Mapping and Potential Haplotypes for unaccounted alleles" as described should produce the promised association with confidence, even across amplicon boundaries. I have observed that in the provided examples analysis the list of Potential Haplotypes is long and no direct measure of (un)certainty is provided. Even if I applaud the ambitious goal to deduce genomic background of unaccounted alleles, since this would be very important applications, I would either like to see a more elaborated evaluation, maybe even in the context of recombination events, or at least to tone down the language a bit.

## minor issues and suggestions

* Fig 2: assuming that the points at x = 0 and y = 0 represent FP and FN, respectively, please elaborate how the different tools perform with respect to correct classification (detection instead of quantification)

* I assume that "All MATs were pre-annotated with PANGO lineage annotations." does only apply to the SARS-CoV-2 tree and not to RSV-A and Mpox trees mentioned before this sentence. If so, please correct and specify the lineage annotation regime used for RSV-A and Mpox.

* "For the samples used to mimic realistic cluster emergence scenarios, we first removed the haplotypes found (via ClusterTracker) in the new region from the MAT and simulated 5% of wastewater reads from one of the removed haplotypes, while the remaining 95% were drawn from lineages circulating in the region at that time.". What does "new region" refer to? Please elaborate.

* "For these experiments, we created four base mixtures and simulated each with a novel haplotype at 5%, 10%, or 20% abundance, resulting in 12 samples with 1.6 million reads each." Please specify the nature of simulated haplotypes (number of alleles, distance between alleles).

* please consider to comment (even if it is just in the discussion) how WEPP would perform in the detection of novel recombinant lineages not present in the MAT used.

* what is the time complexity of the algorithm as a function of tree size and read numbers?

* I did not manage to find the views described in Fig 1B(ii-iv) in the dashboard.

Warm regards, Fabian Amman

Reviewer #2: Gangwar et al. introduce a computational pipeline called WEPP, designed to enhance the resolution and analytical capabilities of wastewater-based epidemiology (WBE). By leveraging phylogenetic placement of sequencing reads onto mutation-annotated trees (MATs), WEPP enables the identification of near-exact viral haplotypes directly from wastewater samples. This approach goes beyond lineage-level detection, allowing for the discovery of novel variants, the inference of multiple introductions, and the early identification of emerging pathogens sometimes weeks before clinical confirmation.

The manuscript states that WEPP "uncovered biological insights missed by other tools" and "detected variants up to five weeks before clinical confirmation," but does not present quantitative performance metrics or confidence intervals. I would to suggest include statistically robust benchmarking with synthetic datasets (such as SWAMPy or Freyja benchmark) and real data tested in parallel with Freyja.

In addition, although WEPP has been applied to various pathogens, there is no explicit analysis of the impact of coverage and read quality, which are highly variable factors in environmental samples. I recommend including simulated experiments with different levels of coverage and baseline quality (Q-score), evaluating the algorithm's behavior. This analysis would demonstrate the method's robustness in low-quality contexts, common in WBE.

The claim that WEPP detects variants up to five weeks before clinical confirmation is highly relevant, but lacks epidemiological context. The inclusion of a time-series table or graph (preferred) comparing WBE detection data with corresponding clinical records should demonstrate the magnitude and consistency of the temporal gain.

How does WEPP account for the impact of recombination events in RNA viruses on phylogenetic placement and downstream haplotype inference? Given that recombination can obscure true evolutionary relationships and lead to misleading branch placements in mutation-annotated trees, it would be important to clarify whether WEPP incorporates any correction, detection, or filtering strategy for recombinant genomes, and how such cases might affect its resolution and interpretation of wastewater-derived sequences.

Reviewer #3: This manuscript by Gangwar et al. describes a pathogen-agnostic pipeline that can improve wastewater based pathogen surveillance. It is based on phylogenetic placement of sequencing reads onto mutation-annotated trees. Most of the validation in this manuscript was conducted using SARS-CoV-2 sequecing data, with validation extended to RSV-A and Mpox data. The tool enables identification of haplotypes, reporting the abundance of haplotypes and lineages as well as can flag potential novel variants. Data are integrated in an interactive dashboard for visualization. All of these features combined make a powerful package for future application. However, as presented the manuscript is overly long, repetitive at times and a more critical assessment of what the tool can and can’t do would be greatly beneficial to the reader.

Specific comments (note: there is a lack of page / line numbers):

1. Summary / last sentence: WWE has been used previously to track lineages and report on variants; please revise that this was previously only restricted to clinical surveillance.

2. Abstract: Statement that WEPP detected findings that previous datasets missed does not seem to be true for all datasets.

3. Introduction: Starting at paragraph 4 (nearly half of the introduction) the introduction is a summary of the results (even including reference to figures). While a couple of sentences summarizing the approach and results are ok, this is excessive (and the manuscript ends up being repetitive).

4. Results (and also in intro): prediction of subsequent cases is indicated as 2 weeks. This needs context in terms of sampling scale of WW collection as well as likely organism, lineage, immune status of the population. Other reports have indicated a 1 week lag time

5. The manuscript refers to high-error ONT sequencing in multiple sections. However, this appears to be an outdated statement given the improvements over the past few years in chemistry and basecalling. Please reword.

6. Repetition: the section on intralineage clusters etc introduces the datasets at the beginning; and then later discusses them; this can be much condensed.

7. The discussion needs a limitations section. This should also include a discussion on the importance of epidemiologic data and sampling scales (features the tool itself cannot address).

**Have the authors made all data and (if applicable) computational code underlying the findings in their manuscript fully available?**

The PLOS Data policy requires authors to make all data and code underlying the findings described in their manuscript fully available without restriction, with rare exception (please refer to the Data Availability Statement in the manuscript PDF file). The data and code should be provided as part of the manuscript or its supporting information, or deposited to a public repository. For example, in addition to summary statistics, the data points behind means, medians and variance measures should be available. If there are restrictions on publicly sharing data or code —e.g. participant privacy or use of data from a third party—those must be specified.requires authors to make all data and code underlying the findings described in their manuscript fully available without restriction, with rare exception (please refer to the Data Availability Statement in the manuscript PDF file). The data and code should be provided as part of the manuscript or its supporting information, or deposited to a public repository. For example, in addition to summary statistics, the data points behind means, medians and variance measures should be available. If there are restrictions on publicly sharing data or code —e.g. participant privacy or use of data from a third party—those must be specified.requires authors to make all data and code underlying the findings described in their manuscript fully available without restriction, with rare exception (please refer to the Data Availability Statement in the manuscript PDF file). The data and code should be provided as part of the manuscript or its supporting information, or deposited to a public repository. For example, in addition to summary statistics, the data points behind means, medians and variance measures should be available. If there are restrictions on publicly sharing data or code —e.g. participant privacy or use of data from a third party—those must be specified.requires authors to make all data and code underlying the findings described in their manuscript fully available without restriction, with rare exception (please refer to the Data Availability Statement in the manuscript PDF file). The data and code should be provided as part of the manuscript or its supporting information, or deposited to a public repository. For example, in addition to summary statistics, the data points behind means, medians and variance measures should be available. If there are restrictions on publicly sharing data or code —e.g. participant privacy or use of data from a third party—those must be specified.

Reviewer #1: Yes

Reviewer #2: Yes

Reviewer #3: Yes

PLOS authors have the option to publish the peer review history of their article (what does this mean?). If published, this will include your full peer review and any attached files.). If published, this will include your full peer review and any attached files.). If published, this will include your full peer review and any attached files.). If published, this will include your full peer review and any attached files.

...

Reviewer #1: **Yes:** Fabian AmmanFabian AmmanFabian AmmanFabian Amman

Reviewer #2: No

Reviewer #3: No

**Figure resubmission:**
---

## [Decision Letter · Decision Letter 1]

13 Mar 2026

Dear Professor Turakhia,

We are pleased to inform you that your manuscript 'WEPP: Phylogenetic Placement Achieves Near-Haplotype Resolution in Wastewater-Based Epidemiology' has been provisionally accepted for publication in PLOS Computational Biology.

Best regards,

Lin Hou

Academic Editor

PLOS Computational Biology

Thomas Leitner

Section Editor

PLOS Computational Biology

Reviewer's Responses to Questions

**Comments to the Authors:**

Reviewer #1: During revision of the manuscript "WEPP: Phylogenetic Placement Achieves Near-Haplotype Resolution in Wastewater-Based Epidemiology" by Pranav Gangwar et al. all issues raised during the initial reviewing were adequately addressed.

The only minor issue I have spotted:

"removing its neighboring haplotypes (indicated by ‘Node Removal Radius’, Table 1). As evident" should read

"removing its neighboring haplotypes (indicated by ‘Node Removal Radius’, Table 2). As evident"

**Have the authors made all data and (if applicable) computational code underlying the findings in their manuscript fully available?**

The PLOS Data policy requires authors to make all data and code underlying the findings described in their manuscript fully available without restriction, with rare exception (please refer to the Data Availability Statement in the manuscript PDF file). The data and code should be provided as part of the manuscript or its supporting information, or deposited to a public repository. For example, in addition to summary statistics, the data points behind means, medians and variance measures should be available. If there are restrictions on publicly sharing data or code —e.g. participant privacy or use of data from a third party—those must be specified.requires authors to make all data and code underlying the findings described in their manuscript fully available without restriction, with rare exception (please refer to the Data Availability Statement in the manuscript PDF file). The data and code should be provided as part of the manuscript or its supporting information, or deposited to a public repository. For example, in addition to summary statistics, the data points behind means, medians and variance measures should be available. If there are restrictions on publicly sharing data or code —e.g. participant privacy or use of data from a third party—those must be specified.requires authors to make all data and code underlying the findings described in their manuscript fully available without restriction, with rare exception (please refer to the Data Availability Statement in the manuscript PDF file). The data and code should be provided as part of the manuscript or its supporting information, or deposited to a public repository. For example, in addition to summary statistics, the data points behind means, medians and variance measures should be available. If there are restrictions on publicly sharing data or code —e.g. participant privacy or use of data from a third party—those must be specified.requires authors to make all data and code underlying the findings described in their manuscript fully available without restriction, with rare exception (please refer to the Data Availability Statement in the manuscript PDF file). The data and code should be provided as part of the manuscript or its supporting information, or deposited to a public repository. For example, in addition to summary statistics, the data points behind means, medians and variance measures should be available. If there are restrictions on publicly sharing data or code —e.g. participant privacy or use of data from a third party—those must be specified.

Reviewer #1: Yes

PLOS authors have the option to publish the peer review history of their article (what does this mean?). If published, this will include your full peer review and any attached files.). If published, this will include your full peer review and any attached files.). If published, this will include your full peer review and any attached files.). If published, this will include your full peer review and any attached files.

...

Reviewer #1: No

---

## [Editor Report · Acceptance letter]

PCOMPBIOL-D-25-01603R1

WEPP: Phylogenetic Placement Achieves Near-Haplotype Resolution in Wastewater-Based Epidemiology

Dear Dr Turakhia,

I am pleased to inform you that your manuscript has been formally accepted for publication in PLOS Computational Biology. Your manuscript is now with our production department and you will be notified of the publication date in due course.

With kind regards,

Anita Estes
